# Linking Weather Regimes to the Variability of Warm-Season Tornado Activity over the United States

Authors: Matthew Graber[1], Zhuo Wang[1], & Robert J. Trapp[1]

[1]Department of Climate, Meteorology, & Atmospheric Sciences, University of Illinois Urbana-Champaign, Urbana, 61820, United States

*Correspondence to*: Zhuo Wang (zhuowang@illinois.edu)

**Abstract**. The contiguous United States (CONUS) experiences considerable variability in tornado activity on seasonal time scales. The high impacts of tornadoes in the CONUS motivate the need to better understand the link between seasonal tornado activity and large-scale atmospheric circulation, which may contribute to better seasonal prediction. We employed K-means clustering analysis of low-pass/EOF filtered 500 hPa geopotential height (500H) daily anomalies from the ERA-5 reanalysis and identified five warm-season weather regimes (WRs). Certain WRs are shown to strongly affect tornado activity, especially outbreaks, due to their relationship with environmental parameters including convective available potential energy (CAPE) and vertical wind shear (VWS). In particular, WR-B, which is characterized by a three-cell wave-like pattern with an anomalous low over the central-CONUS, is associated with enhanced CAPE and VWS in tornado-prone regions and represents a tornado-favorable environment. Persistent WRs, those lasting for ≥5 consecutive days, are associated with 75% of all tornado outbreaks (days with >10 EF-1+ tornadoes) since 1960; persistent WR-B, in particular, accounts for about 31% of all tornado outbreaks. The impacts of WR persistence on tornado activity anomalies, however, are found to be asymmetric: compared to non-persistent WRs, persistent WRs amplify positive tornado activity anomalies but may not further enhance negative tornado activity anomalies. An empirical model using WR frequency and persistence captures the year-to-year variability of warm-season tornado days and outbreaks reasonably well, including some years with high-impact outbreaks. Our study highlights the potential application of WRs for better seasonal prediction of tornado activity.

## 1 Introduction

The contiguous United States (CONUS) experiences more tornadoes than anywhere else in the world, leading to significant economic and life losses (NCEI, 2024). Tornado outbreaks (TOs) are a primary contributor to these impacts, and the annual TO frequency has increased by 2.5 events over the past 63 years (Brooks et al. 2014; Graber et al. 2024), particularly over the Southeast U.S. (Gensini and Brooks, 2018; Graber et al., 2024; Moore, 2018; Moore and DeBoer, 2019). In contrast, tornado days (TDs) have decreased in frequency at a rate of ~10 per decade since 1960 (Brooks et al. 2014; Graber et al. 2024), especially from March to September, and over the southern Great Plains (Gensini and Brooks, 2018; Graber et al., 2024; Moore, 2018; Moore and DeBoer, 2019). Embedded within these trends is large interannual variability, as evidenced by the percent change, with respect to the previous year, in annual CONUS tornado reports over the recent five years (2019-2023): +34.7%, -28.7%, +21.4%, -13.0%, and +24.5%, as well as by the corresponding percent change in tornado fatalities: +320.0%, +80.9%, +36.8%,

-77.9%, and +260.9% (Storm Prediction Center, 2024). Such variability affects the situational
awareness and vulnerability of the populations, especially those that are disadvantaged. It also
complicates decision making and resource management by key stakeholders across multiple
sectors. In addition, exposure to future tornadoes is increasing with growing urban areas (Ashley
and Strader, 2016; Strader et al., 2017, 2024). These and other impacts motivate the efforts to
better understand the variability of tornado activity over the seasonal and longer time scale,
which would ultimately contribute to improved prediction of tornado activity.
Some variability of tornado activity can be attributed to low-frequency climate modes (Miller et
al., 2022; Niloufar et al., 2021; Thompson and Roundy, 1998; Vigaud et al., 2018b). For
example, Cook and Schaefer, (2008) examined winter tornado outbreaks in relation to the phase
of the El Niño – Southern Oscillation (ENSO) and found that a La Niña phase favored tornadoes
in the Southeast and a neutral phase favored tornadoes in the Great Plains. Allen et al., (2015)
further found that La Niña years typically coincide with more tornadoes in the spring and El Niño
years with fewer tornadoes across the central CONUS, and that the winter ENSO phase can be
used to predict tornado frequency during the spring. Additionally, a positive (negative) phase of
the Arctic Oscillation (AO) combined with a La Niña (El Niño) phase may increase (decrease)
tornado activity (Tippett et al., 2022). Tornado activity can also be modulated by anthropogenic
climate change, either indirectly via changes in climate modes or/and directly via changes in
relevant atmospheric conditions. For example, increasing greenhouse gas concentrations are
projected to lead to a moister atmosphere, especially in the lower troposphere, which contributes
to higher convective available potential energy (CAPE)  (Trapp et al., 2007). Diffenbaugh et al.
(2013) showed that climate models project a robust increase in the number of days with high
CAPE coinciding with high vertical wind shear (VWS) in the eastern U.S., two important
parameters for tornado-favorable environments (Brooks et al., 2003; Mercer and Bates, 2019;
Rasmussen and Blanchard, 1998; Thompson et al., 2012).
Variability of synoptic-scale circulations provides another means of explaining tornado activity.
Conducive synoptic-scale circulation anomalies for TOs in the United States show a trough-ridge
pattern over the central- to eastern-CONUS, while non-TOs usually feature more zonal flow
(Mercer et al., 2012). In particular, Cwik et al. (2022) performed rotated EOF analysis of 500-hPa
geopotential height associated with historic May TOs and identified three circulation patterns.
The three circulation patterns are all characterized by a trough feature over the central to eastern
U.S. While their study concludes that the synoptic patterns associated with TOs remain the same
from 1950 to 2019, there is partial variability in the locations of TOs on multidecadal scales.
Additionally, mesoscale processes without strong links to synoptic-scale circulations also affect
tornadoes, especially weak or isolated tornado events (e.g., tornadogenesis in non-supercellular
storm modes associated with mesoscale boundaries; see Wakimoto and Wilson, 1989).
In this study, we will investigate the link between the synoptic-scale circulation and tornado
activity using the concept of weather regimes (WRs). Previous studies suggest that WRs
represent a finite number of equilibrium states of the climate system (Charney and DeVore,

1979; Hannachi et al., 2017; Michelangeli et al., 1995). Their spatial patterns are determined by the internal dynamics of the atmosphere, while their frequencies and persistence may be modulated by climate modes or external forcings (Corti et al., 1999). The WR framework thus has a strong dynamic basis. Different from the EOF analysis, WRs are not required to be orthogonal to each other and can thus more flexibly represent various recurrent synoptic-scale circulation patterns.

WRs have been used to detect changes in regional temperature, wind, and precipitation (Grams et al., 2017, 2020; Robertson and Ghil, 1999; Schaller et al., 2018; Vigaud et al., 2018a). In particular, WRs have been used to investigate sub-seasonal variability of tornado activity, and a skillful WR-based, hybrid model was developed for the sub-seasonal prediction of tornado activity in the month of May (Miller et al. 2020). Lee et al., (2023) applied the year-round WR method (Grams et al., 2017) over North America and defined four year-round WRs. Tippett et al., (2024) identified statistically significant relationships between these year-round WRs and tornado activity in all months except June through August, but Tippett et al. (2024) made no consideration of WR persistency. This helps motivate our focus herein on the warm-season tornado activity and its interannual variability. Additionally, unlike Cwik et al. (2022)'s study, which focuses on circulation patterns conditioned on major TOs, our identification of WRs is independent of TOs. This approach allows us to examine WRs that facilitate or hinder tornado activity, providing more comprehensive information for potential forecasting applications. Furthermore, we will examine environmental conditions relevant to tornado development, such as CAPE and VWS (Brooks et al., 2003; Mercer and Bates, 2019; Rasmussen and Blanchard, 1998; Thompson et al., 2012), which will help us better understand the link between WRs and tornado activity.

The remainder of the paper is organized as follows. First, a detailed overview of the methodology for WR identification and empirical model development is presented. The links between WRs and tornado activity is then shown through composite anomalies of CAPE and VWS, as well as through anomalies of precipitation and TD probability. Next, WR persistence is analyzed to test the hypothesis that persistent WRs are more likely to produce TDs and TOs. Finally, the WR components, frequency and persistence, are incorporated into an empirical model to evaluate the potential of WRs to improve seasonal tornado prediction. The paper culminates in a discussion of the possible applications of our results.

## 2 Data and Methodology

### 2.1 ERA-5 reanalysis data

Data from the ERA-5 reanalysis (Hersbach et al., 2020) were analyzed over the CONUS $[24 - 55^o \text{ N}, 130 - 60^o \text{ W}]$ at the native $0.25^o$ latitude $\times 0.25^o$ longitude resolution. This includes daily mean 500 hPa geopotential heights (500H). Daily maximum values of most unstable CAPE (MUCAPE) and convective precipitation (CP) were used to represent the daily peak instability. The 0-6 km bulk wind shear (S06), or deep-layer shear, was estimated at 3-h

intervals as the magnitude of the vector difference between the 500 hPa and 10 m winds. The
daily mean S06 was then calculated at each ERA-5 grid point. Daily anomalies of a variable
were defined with respect to the daily climatology on every calendar day, following:

$$H'(d, y) = H(d, y) - \bar{H}(d) \tag{1}$$

where $y$ is year, $d$ is calendar day, $H$ is the variable or parameter of consideration, and the
overbar denotes the long-term mean. A 2° (latitude) x 2° (longitude) uniform filter was applied to
MUCAPE, CP, and S06 anomalies to coarsen the data and were tested for significance using a
one-sample, two-sided t-test.
Following Graber et al. (2024), all analyses were conducted over the period 1960-2022, and
focused specifically on the warm season, defined as April to July (AMJJ), which is peak season
of tornado activity in the United States.

## 2.2 Weather Regimes

To identify weather regimes, daily anomalies of 500H at each grid-point were first detrended by
removing the linear trend of the seasonal mean (AMJJ) 500H averaged over the entire Northern
Hemisphere (Fig. S1). The detrending approach removed the positive trend of hemispheric mean
500H while preserving the spatial patterns and potential changes of WR frequency or
persistence. Although geopotential height anomalies were normalized prior to the K-means
clustering in the year-round WR analysis by Tippett et al. (2024) and Lee et al. (2023), 500H
anomalies are not normalized in this study because we focus on one season, which is consistent
with many previous studies (Miller et al., 2020; Robertson and Ghil, 1999; Vigaud et al., 2018a;
Zhang et al., 2024). A 5-day low-pass filter was applied to 500H anomalies, and the leading 8
EOFs, accounting for ~90% of the variance, were retained in the EOF dimension reduction; such
pre-processing does not qualitatively affect the regime patterns or the regime frequencies (Figs.
S2-S4) but does facilitate comparison with previous studies. K-means clustering analysis was
applied to the 500H daily anomalies over the CONUS, and the number of clusters was
determined as five (Fig S5-S6) using the elbow method and, more clearly, the Davies-Bouldin
Index (Davies and Bouldin, 1979; Kodinariya and Makwana, 2013; Miller et al., 2020), which is
consistent with Zhang et al. (2024). A persistent WR was defined as a WR lasting for $\geq 5$
consecutive days.

## 2.3 Tornado Reports

Tornado reports for the period 1960-2022 were obtained from the NOAA Storm Prediction
Center Severe Weather Database. These reports are georeferenced with time, date, and EF/F
rating. TDs were defined as any day with $\geq 1$ EF/F-1+ tornadoes, and TOs were defined as any
day with >10 EF/F-1+ tornadoes. The >10 threshold provides a larger sample size than higher
thresholds but has a similar trend as >20 or >30 thresholds (Graber et al. 2024). EF/F-0 reports
were not included due to their reporting uncertainty (Brooks et al., 2014; Trapp, 2013).
Nevertheless, there are remaining and well-known biases in this dataset, which we attempt to
manage with a focus on days with tornadoes rather than tornado counts (e.g., Brooks et al. 2014;
Graber et al. 2024; Trapp, 2014).
The TD probability anomalies ($P_{a,i}$) were calculated at each grid-point for each WR as follows:

$$P_{a,i} = \frac{P_i - P_c}{P_c} \times 100 \tag{2}$$

where $P_c$ is the climatological mean TD probability and was calculated as the total number of
TDs divided by the total number of days in the warm-season from 1960-2022, and $P_i$ represents
the TD probability for WR-i (i.e., the number of TDs for WR-i divided by the total WR-i days).
WRs that are (are not as) conducive for TDs would have probabilities above (below) the
climatological mean and thus positive (negative) probability anomalies. The probability
anomalies of TOs and the probability anomalies associated with persistent and non-persistent
WRs were calculated similarly. A Monte Carlo simulation test with 10000 resamples was used to
test for significance of the anomalies. The number of WR-i days was multiplied by the
climatological mean TD probability to get an expected number of tornado days. The p-value was
calculated based on the proportion of simulations that were more extreme than the observations.
**2.4 Empirical model for tornado activity**
Using WR frequency and tornado probabilities for both persistent (subscript *p)* and non-
persistent (subscript *np*) WRs, we developed an empirical model to assess the relationship
between the variability of seasonal tornado activity and WRs:

$$TI(t) = \sum_{i=1}^{5} f(i,t)_p \times P_{i,p} + \sum_{i=1}^{5} f(i,t)_{np} \times P_{i,np} \tag{3}$$

where TI(t) denotes a tornado index for year t. The model takes the count of WR-i days (**f (i, t)**)
for year t and multiplies it by the tornado probability corresponding to WR-i ($P_i$). The WR count
is a function of regime (i) and year (t). The WR tornado probability is only a function of regime
(i) and represents the likelihood that a TD will occur. Probabilities are assessed for persistent and
non-persistent WRs separately, under the hypothesis that persistent WRs contribute to stronger
TD or TO anomalies (Miller et al., 2020; Trapp, 2014). Spearman Rank correlation is used to
compare the modeled and observed tornado indices.
**3  Weather Regimes and Tornado Activity**
The composite mean 500H anomalies for each WR are shown in Fig. 1, ordered with decreasing
frequency of occurrence. WR-A is the most frequent regime and is characterized by an
anomalous high over the west-central-CONUS and a weak anomalous low over the Southeast.
WR-B and WR-C are both characterized by a three-cell wave pattern, with negative and positive
500H anomalies over the central-CONUS, respectively. WR-D and WR-E are west-east dipole

patterns that nearly mirror each other. The WR spatial structures closely resemble the WRs in Zhang et al. (2024) and Miller et al. (2020). WR-A is also similar to the Alaskan ridge pattern in Lee et al., (2023) and Tippett et al., (2024), and WR-D is similar to their Pacific ridge pattern. However, since our study focuses on a different region and a specific season, and is based on a different number of clusters, there are noticeable differences. In particular, the WRs in Lee et al. (2023) have a stronger loading in higher latitudes, probably partly because they normalized geopotential height anomalies by the area-averaged standard deviations with a cosine-latitude weighting, a procedure we choose to exclude.

The potential links between these WRs and tornado activity are indicated by MUCAPE and S06 (Fig. 1a-e). Composite anomalies of these parameters were calculated by subtracting the corresponding climatological mean (Fig. 1f) from the composite mean associated with each WR. The high values of the climatological MUCAPE across the central- and southeastern CONUS are connected to the physical geography of North America (Brooks et al., 2003; Trapp, 2013) and the warm-season climatological mean 850-hPa circulation, with southerly winds transporting heat and moisture into the central-CONUS (Mercer and Bates, 2019). The climatological S06 is characterized by high values over the eastern-CONUS, which are tied to the midlatitude jet stream. With an anomalous 500H high over the west-central-CONUS and an anomalous 500H low over the Southeast, WR-A favors anomalously low MUCAPE and S06 relative to climatological means. In contrast, the anomalous 500H low over central North America and the anomalous 500H high over the southeastern U.S. in WR-B imply enhanced westerly flow and increased moisture and warm-air transport from the Gulf, leading to positive S06 and MUCAPE anomalies in southeastern U.S. The favorable anomalies presented in WR-B agree with the Pacific Ridge findings in Tippett et al. (2024). In WR-C, the anomalous 500H low over western North America and the anomalous 500H high over central North America imply enhanced southerly flow and increased moisture and heat transport leading to positive MUCAPE anomalies in the central U.S., which overlap with reduced S06 south of the anomalous high. For WR-D, the anomalous 500H high over the eastern CONUS and the anomalous 500H low over the western CONUS imply enhanced southerly flow and increased moisture and heat transport, consistent with positive MUCAPE anomalies in the central U.S., while S06 decreases in the south of the anomalous high and increases in the north. WR-E, in contrast to WR-D, implies reduced southerly flow and decreased moisture and warm-air transport and is associated with negative MUCAPE anomalies in the central U.S., but S06 increases substantially over the Southeast.

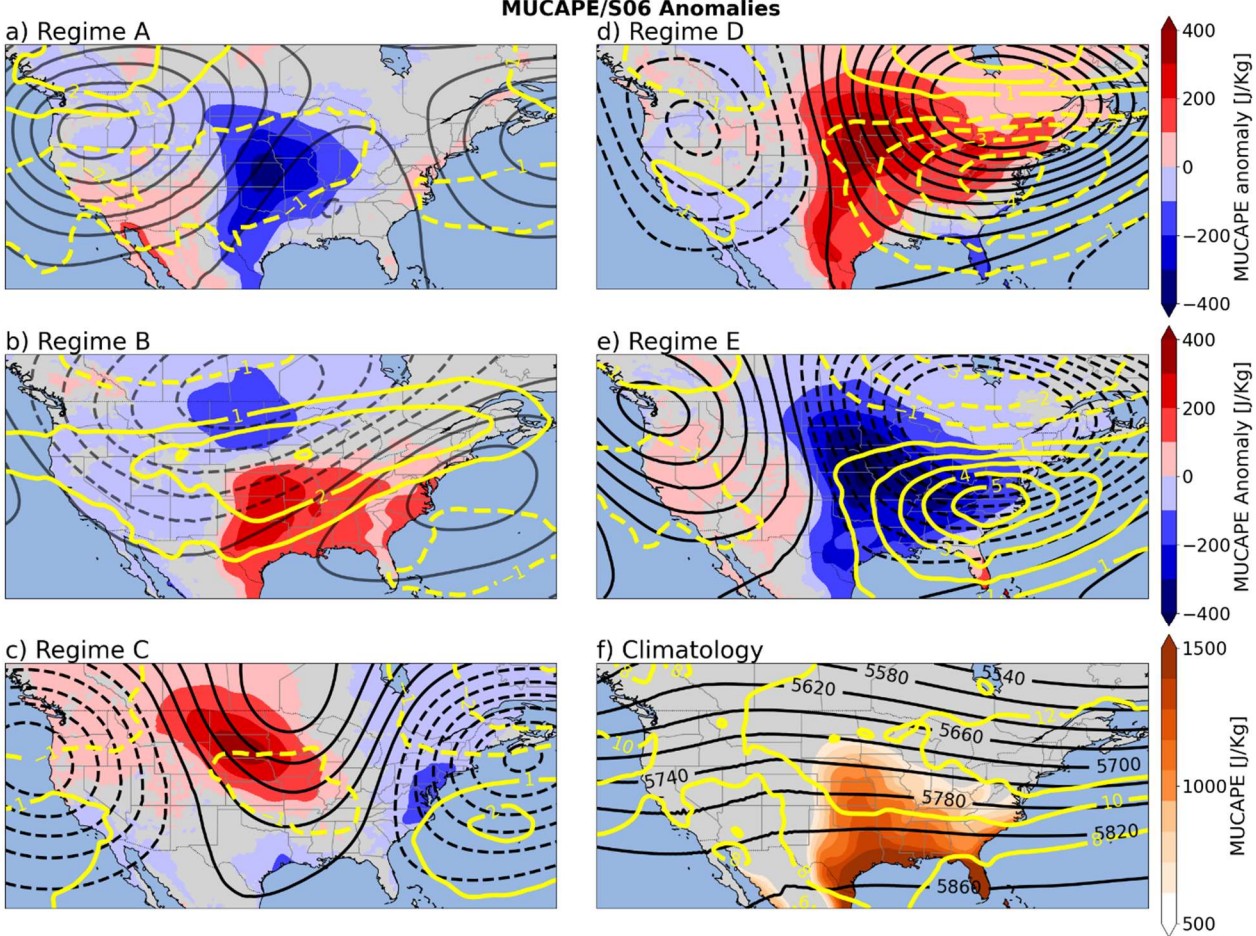

**Figure 1:** (a-e) Composite anomalies of 500H (black contours, +/- 10 m) for each warm-season WR and corresponding 2º x 2º uniform-filtered anomalies of daily maximum MUCAPE (units: J Kg$^{-1}$; color fill) and daily mean S06 (units: m s$^{-1}$; yellow contours). Significance (p<0.05) was tested at each grid point using a one-sample, two-sided t-test to mask out non-significant anomalies; (f) Warm-season climatology of 500H (black contours, labeled), daily maximum MUCAPE (color fill) and daily mean S06 (yellow contours).

The WR-tornado activity link is illustrated by the composite anomalies of TD probability and CP for each WR (Fig. 2a-e). The climatological TD probability and CP are also shown (Fig. 2f) for reference. Here TD probability anomalies are evaluated following Eq. 2 with respect to $P_c$ at each grid point and then smoothed using a scipy gaussian filter with sigma 6. Such smoothing has removed some small-scale anomalies but retained the large-scale patterns. Convective-storm occurrences can be approximated using CP. Convective storms are a necessary but insufficient condition for tornadoes, so more CP does not necessarily lead to more tornadoes, but less CP likely means reduced tornado activity (Tippett et al., 2014). CP anomalies collocate well with the MUCAPE anomalies (Fig. 1) since non-zero CAPE is generally necessary for deep convection, but CP also includes information about convection initiation. WR-A has negative TD anomalies in the central-CONUS, where negative anomalies in CP and MUCAPE/S06 are also present. Positive TD anomalies in the Southeast and Midwest of WR-B are collocated with positive anomalies in CP, MUCAPE, and S06. Despite the negative S06 anomalies, positive TD

anomalies occur in the central Great Plains in a region of positive MUCAPE and CP anomalies
for WR-C. Weak, negative TD anomalies in association with WR-C are found in the Southeast
where negative CP anomalies are present. Positive (negative) TD anomalies in WR-D over the
central-CONUS are collocated with positive (negative) CP and MUCAPE anomalies, and
reduced S06 (Fig. 1d) also contributes to the negative TD anomalies in the Southeast. Finally,
negative TD anomalies occur in the central-CONUS, collocated with negative anomalies of
MUCAPE and CP associated with WR-E, while positive TD anomalies occur in the Southeast
despite reduced MUCAPE. The latter can probably be attributed to the strong positive anomalies
in S06 (Figs. 1e and 2e). However, given the low climatological TD probability in the Southeast
(Fig. 2f), the absolute changes in TD days may not be high. Overall, the distribution of TD
anomalies shows a good agreement with CP and MUCAPE anomalies of the same sign, and S06
seems to play a secondary role in most regions.

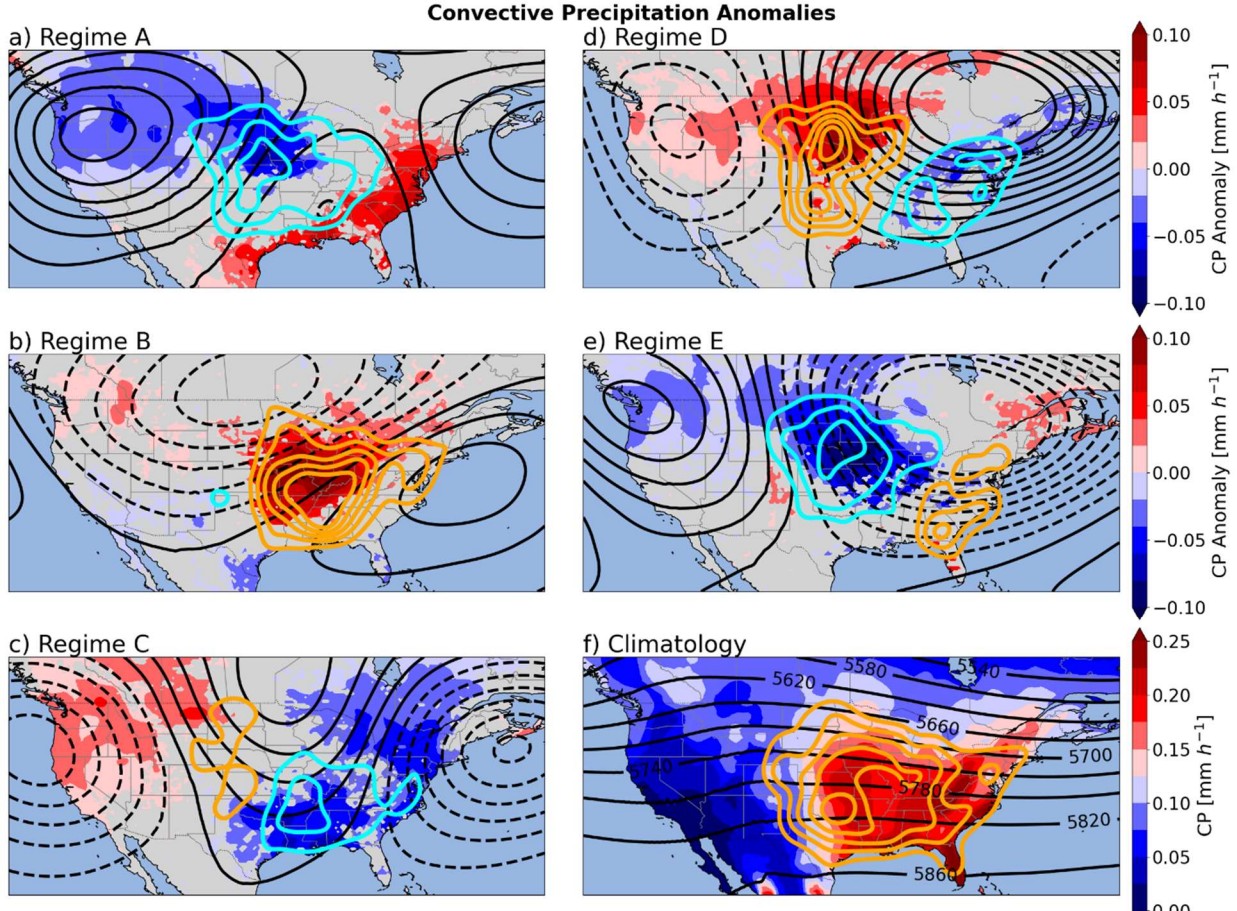


**Figure 2:** a-e): 2° x 2° uniform-filtered, significant composite anomalies of daily maximum CP rate (units: mm h⁻¹; shading) at each grid point (red/blue filled contours) for each WR A-E days with gaussian smoothed TD probability anomalies (contour intervals: +/- 20 %; orange and cyan colors represent positive and negative values, respectively) and 500H for each WR (black contours: +/- 10 m). Significance (p<0.05) is tested at each grid point using a one-sample, two-sided t-test to mask out non-significant anomalies; (f) Climatology of daily maximum CP (shading),

500H (black contours), and tornado day probability (orange contours). Climatological probabilities are shown at
each grid point with contour intervals of +0.01.
The link between WRs and geospatially aggregated tornado activity is summarized in Fig. 3 for
different regions. There are a total of 4348 warm-season TDs from 1960-2022, therefore $P_c \approx$
56.5%. TD probability is enhanced for WR-B and WR-D, with probability anomalies of **+19.2%**
and **+9.8%**, respectively (corresponding to TD probabilities of **67.6%** and **62.3%**; Fig. 3). For
reference, these are associated with large positive TD probability anomalies in the Southeast and
central-CONUS, respectively (Fig. 2). TD probability is reduced for WR-A and WR-E,
associated with negative TD anomalies across the central-CONUS (Fig. 2). The TD probability
anomaly associated with WR-C is close to zero (Fig. 3), which can be attributed to the near-
cancellation between the opposite anomalies in the Southeast and central-CONUS (Fig. 2).
There are 415 warm-season TOs from 1960-2022, therefore $P_c = 5.4\%$. In general, the TO
probabilities have a stronger signal than TDs (yellow bars in Fig. 3). For TOs, WR-A has the
strongest negative signal with a **-60.77%** anomaly while WR-B has the strongest positive signal
with a **+76.39%** anomaly (Fig. 3).  The TO anomalies are consistent with the analysis in Figs. 1-
2, in which WR-A (WR-B) showed reduced (enhanced) MUCAPE, S06, and CP over the central
and Southeast CONUS. WR-D has a positive, significant anomaly of TO probability (**+37.4%;**
Fig. 3), which is consistent with enhanced MUCAPE and CP over the central-CONUS (Figs.
1d&2d). WR-E is associated with a negative TO probability anomaly, which can be linked to the
reduced tornado activity over the central-CONUS (Fig. 2). Further analysis reveals that roughly
**64%** of all TOs occur during a WR-B or D.
We also checked TOs using >20 and >30 tornadoes thresholds (red and purple bars in Fig. 3).
The analysis based on the >20 threshold yields similar results as that defined based on the >10
threshold. Although WR-A and WR-B demonstrate significant and consistent signals for days
with >30 tornadoes, the other WRs exhibit contrasting signals for the >10 and >30 thresholds.
This could be due to the small sample size of TOs when using the >30 threshold, and those
anomalies are not significant.

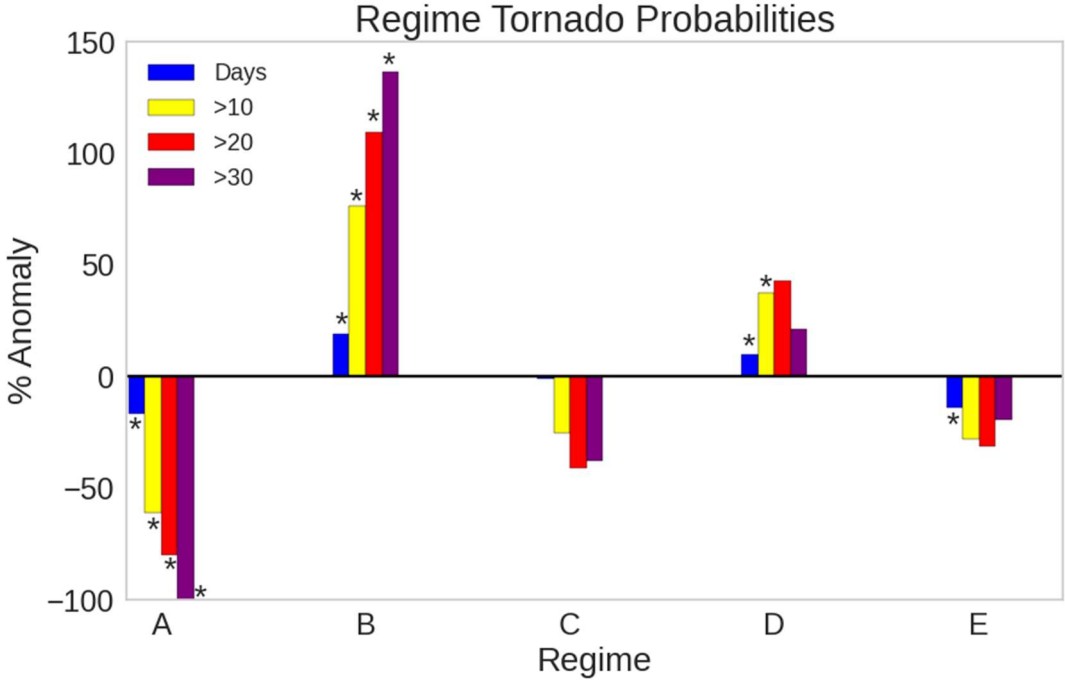


**Figure 3**: Tornado probability anomalies for days with > 0, > 10, > 20, & > 30 tornadoes for each WR in the
CONUS (see Eq. 2 and the related discussion). Anomalies above the 95 % confidence level based on the Monte
Carlo testing (with 10,000 resampling) are regarded as significantly different from zero and marked with an asterisk.

Next we compare persistent and non-persistent WRs to test the hypothesis that persistent regimes
amplify the TD/TO probability anomalies. Persistent WRs are defined as those lasting for at least
5 days. The comparison of the TD probability anomalies between persistent and non-persistent
WRs (Fig. 4a) does not fully support our hypothesis. Although persistent WR-B and WR-D are
associated with a stronger positive anomaly in TD probability than their non-persistent
counterparts, the negative TD probability anomalies are about the same for persistent and non-
persistent WR-A, and persistent WR-E shows an even weaker decrease in TD probability than
non-persistent WR-E. Persistent and non-persistent WR-C show TD probability anomalies of
opposite signs, both with a small magnitude. Compared to TD probability, the anomalies of TO
probability are generally stronger for both persistent WRs and non-persistent WRs. A persistent
WR-B is associated with a TO probability anomaly of ~**90**% and accounts for ~30% of all TOs,
while persistent WR-A is associated with a negative TO probability anomaly of ~**70%**.
However, a consistent picture emerges: persistent WRs amplify the positive anomalies but do not
necessarily enhance the negative anomalies in comparison to the corresponding non-persistent
WRs.

The asymmetric impacts of WR persistence on positive and negative tornado activity anomalies
are also illustrated in Fig. S7. One possible interpretation is that tornado activity indices are
positively defined metrics so they cannot be reduced much further when already close to zero.
However, the results should also be interpreted with caution given the limited sample sizes for
certain groups (Table S1).

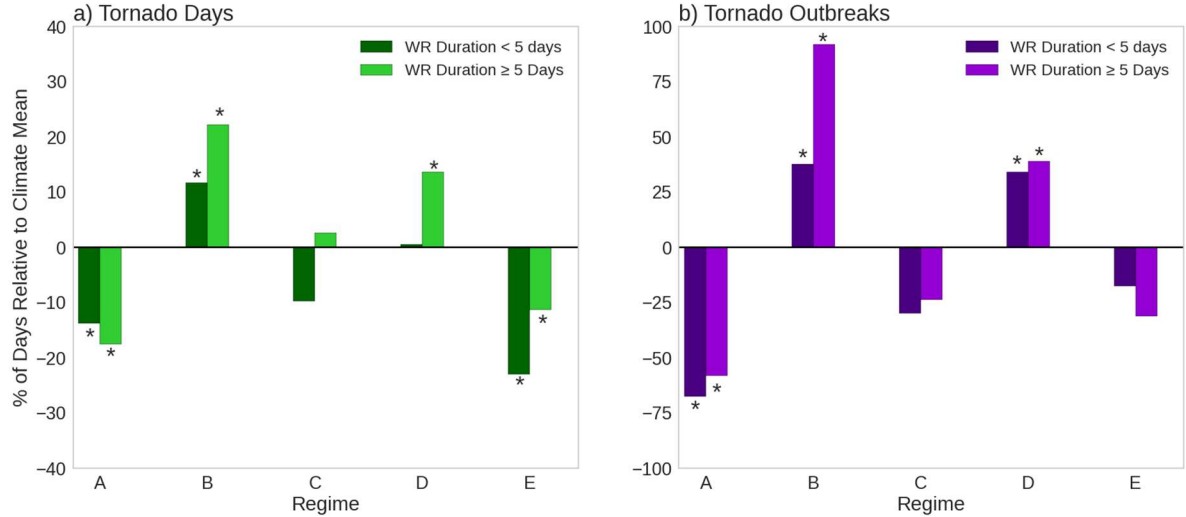

306

**Figure 4: (**a) TD and (b) TO (days with > 10 EF-1+ tornadoes) probability anomalies for each WR for persistent and non-persistent days. Anomalies above the 95 % confidence based on the Monte Carlo testing are marked with an asterisk. Anomalies above the 95 % confidence level based on the Monte Carlo testing (with 10,000 resampling) are regarded as significant and marked with an asterisk.

## 4    Variability of WRs and Tornado Activity

In this section, we further quantify the link between WRs and tornado activity. WR frequencies demonstrate strong interannual and decadal variability (Fig. S8a-e). In particular, WR-A exhibits a frequency increase during the 1980s coinciding with  the steepest decrease in TDs (Brooks et al., 2014; Graber et al., 2024). The increase in seasonal frequency in WR-A is consistent with the spatially similar ridge-trough-ridge WR in Zhang et al., (2024). The frequencies of persistent WRs also show changes across different multidecadal time periods (Fig. S8f).

To examine whether WRs can help explain the interannual and decadal variability of tornado activity over the period 1960-2022, an empirical model was developed following Eq. 3. Figure 5a shows the empirically modelled TDs along with the observed TDs. Despite the decadal variability of WR and persistent WR frequencies (Fig. S8), the empirical model fails to capture the observed decreasing trend or the decadal shift in the 1980s. After detrending the observations using the least-squares fit, the model reasonably represents the interannual variability of TDs (Fig. 5b), with a rank correlation of **0.31** (p-value ~0.01). An empirical framework for EF-3+ TDs was also tested, yielding a rank correlation of **0.41** (Fig. S9). It is interesting to note that the modelled TDs are nearly out of phase with observations in the 1960s, when tornado reports are less reliable (Trapp, 2013). After excluding these years, we reconstructed the empirical model using updated TD probabilities during 1970-2022, and the correlation increases to **0.42** (Fig. S10a)**.** The empirical framework was also tested for EF-3+ TDs during 1970-2022, and the correlation is **0.53** (Fig. S10b).

We also examined TOs. The TO time series from the empirical model has a significant rank
correlation (above the 95% confidence level) with the observations but it underestimates the
observed variance. Since the observed TOs do not have a strong trend, detrending the data does
not affect the results appreciably. Similar to the TD model results (Fig. 5a, b), the TO model is
nearly out of phase with the observations in 1960s. After excluding the data in 1960s, the
correlation between the empirical model and observation increases to **0.38** from 1970-2022 (Fig.
S10c).

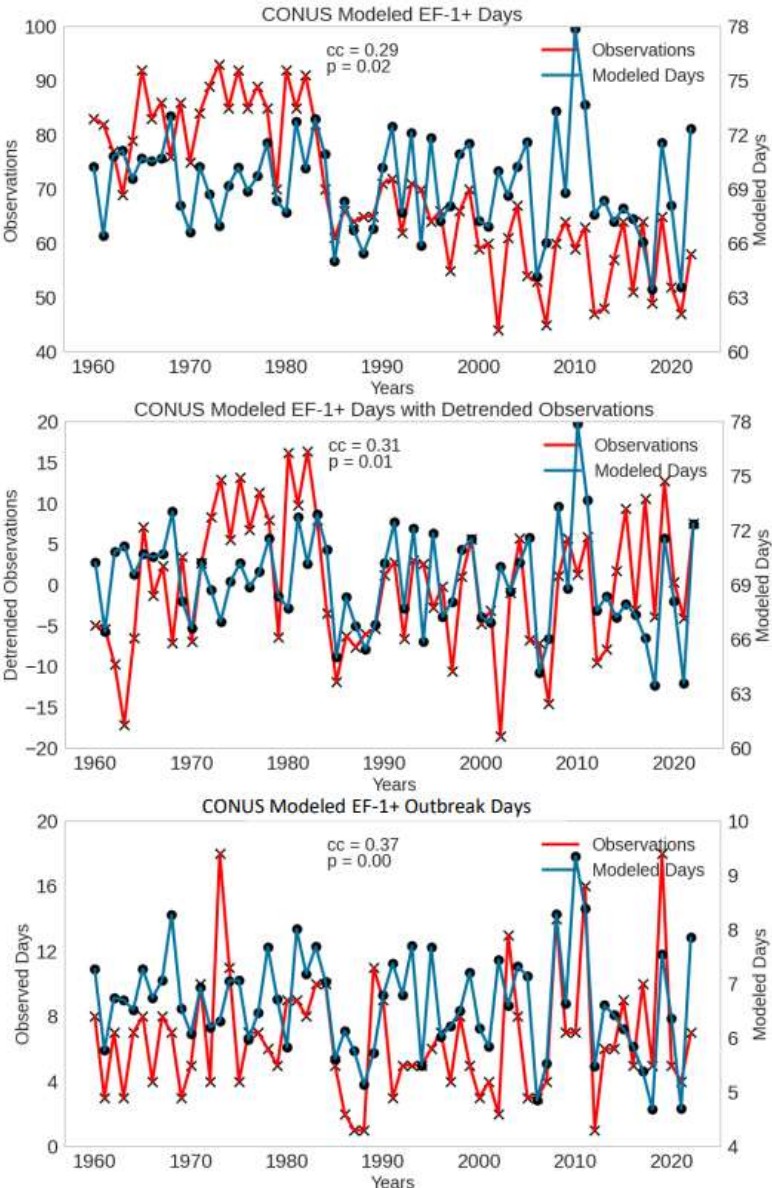


**Figure 5:** Empirically modeled TDs (blue with circles) per year overlaid with (a) observed TDs (red with crosses)
and (b) detrended observed TDs (red with crosses) with spearman rank correlation coefficient (cc) and p-value; (c)
empirically modeled (blue with circles) and observed (red with crosses) TOs per year with the spearman rank
correlation coefficient and p-value.
It is worth noting that although the empirical model captures the interannual variability of TDs
reasonably well, it misses the negative trend or decadal variability of TDs. The empirical model
is constructed under the assumption that probability anomalies of tornado activity associated
with the WRs do not change during the period of analysis. This assumption, however, may not
be strictly valid. For example, significant increases in MUCAPE from P1 to P3 are found for all
five WRs, although S06 undergoes smaller changes (Fig. S11). Additionally, convective
inhibition (CIN) increases in the Southeast and Midwest for WR-B (Fig. S11h) and in the
central-CONUS for WR-C (Fig. S11i) from P1 to P3. Further analysis reveals a lower TD
probability for all WRs in P3 than in P1 (not shown), consistent with the negative trend of TDs
(Graber et al., 2024). A better understanding of dynamic and thermodynamic anomalies
associated with WRs and the role of internal climate variability and anthropogenic forcing in
modulating WRs will help us better understand tornado activity on the decadal and longer time
scales.

**5. Summary**
The weather regime concept was used to investigate the link between synoptic-scale circulation
patterns during the warm season and the variability of corresponding tornado activity over the
U.S. on the interannual time scale. Five WRs were identified over North America using the K-
means clustering analysis of daily mean 500H anomalies from the ERA-5 reanalysis. WR-A is a
three-cell wave-pattern and is associated with negative anomalies of tornado activity in the
central-CONUS, which is consistent with negative anomalies of MUCAPE and S06 over that
region. WR-B is a three-cell wave-pattern that contributes to increased tornado activity in the
Southeast as evidenced by positive anomalies in MUCAPE and S06 there. WR-C is a three-cell
wave-pattern with negative 500H anomalies over both coasts. It is associated with positive
MUCAPE anomalies over the central-CONUS and negative S06 anomalies. It exhibits
climatologically average tornado activity, but it does make a positive, spatially small,
contribution to tornado activity in the Great Plains. WR-D and WR-E are both dipole patterns
with positive and negative 500H anomalies over the east coast, respectively. WR-D contributes
to anomalously positive tornado activity in the Great Plains while WR-E contributes to
anomalously negative tornado activity in the Great Plains. WR-E also contributes to positive
anomalies of tornado activity in the Southeast. A year that includes a high number of WR-B days
is likely to have an above average number of TDs and TOs. In contrast, a year with a high
number of WR-A days would likely have a below average number of TDs and TOs
We tested the hypothesis that WR persistence amplifies the tornado activity anomalies,
regardless of positive or negative anomalies. However, the impacts of WR persistence on
positive and negative tornado activity anomalies are found to be asymmetric: persistent WRs
amplify the positive anomalies but may not further enhance the negative anomalies. This can
probably be attributed to the positive-definite nature of tornado activity indices. While persistent
WRs with favorable environmental conditions (such as WRs B and D) may further increase
tornado activity, TD or TO probability cannot be reduced much further by the persistence of a
tornado-unfavorable WR (such as WR-A) when they are already close to zero.

Using WR frequency and persistence, an empirical model was developed to quantify the relationship between tornado activity and warm-season WRs. The empirical model skillfully estimated the interannual variability of tornado days or TO days, and the model performance was better after excluding the data in the 1960s. Since the empirical model used WR frequency derived from the ERA reanalysis, its predictive skill can be considered an upper bound for this empirical prediction framework, assuming the perfect knowledge of WR frequencies. The atmospheric general circulation model simulations by Straus et al. (2007), which were forced by observed SST and sea ice, suggested the predictability of WRs, thus indicating the potential value of this approach.

The empirical model, however, misses the trend or the multi-decadal variability of TDs. This model deficiency could be attributed to the non-stationary relationship between WRs and tornado activity on the multi-decadal time scale, which is illustrated by the increase in CAPE for all WRs in the more recent decades. The roles of internal variability and anthropogenic forcing, however, are outside the scope of the present study and merit further investigation. Furthermore, although not explored in this study, WRs and tornado activity may both be modulated by large-scale, low-frequency climate modes (Cook and Schaefer, 2008; Lee et al., 2023; Niloufar et al., 2021; Tippett et al., 2024; Vigaud et al., 2018a). Given the potential predictability of WRs (Straus et al., 2007), they may act as an intermediary between large-scale climate modes and tornado activity, while the low-frequency modes may be important sources of predictability for the interannual variability of tornado activity. Overall, weather regimes offer a promising path for developing skillful seasonal tornado prediction models. Such efforts are ongoing and will be reported in due course.

**Code Availability**

Weather Regime identification code and WR structures/labels are available at:

https://github.com/Matt0604/Kmeans

**Data Availability**

The ERA5 data are available at the NCAR research data archive (RDA) (d633000) and Copernicus Climate Data Store (CDS).

doi: 10.5065/BH6N-5N20

The tornado report data used in this study are available through the NOAA Storm Prediction Center severe weather database.

https://www.spc.noaa.gov/wcm/#data

**Author Contributions**

Conceptualization: ZW, RJT, MG

Methodology: MG, ZW, RJT

Project Administration: RJT, ZW

Supervision: ZW, RJT

Writing – Original Draft: MG

Writing – Review and Edits: MG, ZW, RJT

**Competing Interests:**

Authors declare they have no competing interest.

**Acknowledgements**

We acknowledge the NCAR Computation and Information Systems Laboratory (CISL) for providing computing resources through Derecho. All ERA5 data are available used in this study are available at the research data archive. Tornado report data is available through the NOAA Storm Prediction Center severe weather database.

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
