# Peer review of "Activity over the United States"

_EGUsphere, 2024_

## Author Comment (AC1)

Manuscript ID: egusphere-2024-3216

*We appreciate the comprehensive and constructive questions and suggestions from the reviewers. Below are our responses in blue italics. Any necessary revisions made to our manuscript are also indicated below.*

Reviewer # 1:

The regime classification methodology differs from standard ones (EOFs are not used, there is no time filtering, once per day hourly snapshots of 500 hPa heights are used, the choice of cluster number is subjective, there is no normalization of variance), and there is no indication how these methodological choices impact the results. Variance normalization is important because 500 hPa height anomalies have substantially greater variance in April than in July. The k-means clustering method minimizes within-cluster variance, and seasonality in variance might bias the results. Consequently, it may well be the case that the weather regime frequencies vary with month (climatologically), which would confound any analysis with tornado frequency whose climatology also varies by month. Whether this is case or not is unclear because diagnostics such as the seasonality of regime frequency, variance explained, association with modes of large-scale variability, etc. are missing.

*We thank the reviewer for raising this important question. "There is no unique or optimal way of classifying weather regimes" (Robertson and Ghil 1999). In particular, Falkena et al. 2020 (https://rmets.onlinelibrary.wiley.com/doi/10.1002/qj.3818) argued against the use of either EOFs or time filtering on top of K-means clustering because K-means clustering reduces the dimensions and temporal filtering changes the frequencies of weather regimes. In addition, our analysis shows that the application of the 5-day running mean or EOF dimension reduction prior to K-means does not qualitatively affect the regime patterns or the regime frequencies (Figs. R1-R3). We thus chose to use the simplest procedures for regime classification.*

*Whether the data should be normalized is an interesting question for debate. We choose not to normalize the data for a couple of reasons. First, our weather regime analysis covers only four months, and seasonality is thus not as much of an issue, especially since we have removed the seasonal cycle (defined as the long-term daily mean on each calendar day). Second, our focus is on the link between weather regimes and tornado activity, which is quantified much differently than weather regimes. Tornado activity is not normalized for the same reason, given that we are working in one season where tornado activity is relatively common throughout the season, seasonality is not much of an issue. Since we do not normalize tornado activity (and the associated environmental parameters), we believe it is better not to normalize H500 for consistency.*

*"While many previous studies applied a low-pass filter or/and EOF dimension reduction prior to K-means clustering analysis (e.g., Robertson et al., 2020, Lee et al. 2023), Falkena et al. 2020 cautioned against the use of either EOFs or time filtering on top of K-means clustering. Our analysis shows that the application of the 5-day running mean or EOF dimension reduction prior*

to K-means does not qualitatively affect the regime patterns or the regime frequencies (Figs. R1-R3). We thus chose to use the simplest procedures for regime classification."

*The choice of k in K-means clustering is often somewhat subjective, because a metric does not always indicate an unambiguous optimal cluster number, and different metrics may yield different optimal cluster numbers (Dorrington and Strommen 2020, https://doi.org/10.1029/2020GL087907). This is a known limitation of K-means. We tried k=4 and 5. With k=4, 3 of the 4 regimes in our analysis are similar to those in Lee et al. (2023): WR-D and their Greenland High, WR-C and their Alaskan Ridge, and WR-A and their Pacific Trough, but it misses WR B in the k = 5 analysis, which is spatially similar to WR-A in Miller et al. 2020 and the Pacific Ridge in Lee et al. 2023 and is most favorable to tornado activity. We thus chose k=5 to incorporate this important regime.*

*Though some spatial similarities exist, there are some differences as well, which is not unexpected given our focus on one season. These similarities and differences have been further discussed in our study. Additionally, it is worth pointing out that the optimal number of WRs is higher likely because we used the total 500H anomalies and did not apply the EOF dimension reduction. (Falkena et al. 2020).*

"Some WRs are similar to the year-round WRs in Lee et al., (2023), which were subsequently used by *Tippett et al. (2024)*. More specifically WR-A features spatial similarities to a Pacific Trough, WR-B and WR-D show warm and cool phases of a Pacific Ridge and WR-E is characterized by an Alaskan Ridge. WR-C features spatial similarities to a Greenland High as well. It is worth mentioning our study focuses on a different region, a different season and chooses a different k value, and there are thus noticeable differences. WR-A features two anomalous highs over the two coasts as opposed to one anomalous high over the central-CONUS. The anomalous low in WR-B is more pronounced than in Lee et al., (2023). The anomalous high in WR-C is wavelike unlike the Greenland high in Lee et al., (2023). The dipoles in WR-E are further south than they are in the Alaskan Ridge in Lee et al., (2023)."

*The 'once-per-day snapshot' approach was pursued because the chosen time (2100 UTC) of 500H represents a typical time of day when U.S. tornado outbreaks are ongoing (Cwik et al. 2022), thus potentially providing a more straightforward connection for WRs to serve as an intermediate between climate change and tornado activity.*

*Note, Figures R1-R3 have been added into the supplemental information as Figures S2-S4. The manuscript has been updated accordingly.*

[Figure]

**Figure R1:** Original 500H weather regimes in the manuscript.

ERA-5 AMJJ 1960-2022: Low-Pass Filtered

[Figure]

**Figure R2:** 500H weather regimes created applying a 5-day low-pass filter prior to K-means clustering analysis.

ERA-5 AMJJ 1960-2022

[Figure]

**Figure R3:** As in Fig. R1 except created using the first 5eofs for K-means clustering analysis and ordered by spatial structure as in Fig. R1 & R2.

ERA-5 AMJJ 1960-2022

[Figure]

**Figure R4:** As in Fig. R1 except created for k = 4 clusters.

The new regime classification is not compared with previous ones from the same authors for April and May and with year-round regime classifications from Lee et al., (2023) [data is in Zenodo for download]. Making connections to previous work would increase the value of the current work. The classification data (data needed to classify independent data and classification of the days in the study) should be provided.

*In the revised version of the manuscript, we now compare to more previous studies such as Lee et al. 2023, Robertson and Ghil 1999 for example. Lines 162-165 already make connections to Lee et al. 2023.*

*Weather regime identification code is available on github at:*

*https://github.com/Matt0604/Kmeans*

*The ERA-5 reanalysis data and the tornado report data are both publicly available.*

*ERA-5:* https://doi.org/10.24381/cds.bd0915c6

*Tornado report:* https://www.spc.noaa.gov/wcm/#data

*"The WR framework thus has a strong dynamic basis and have been used to reliably detect changes in regional temperature and precipitation previously (e.g., Robertson and Ghil, 1999)."*

"Some WRs are similar to the year-round WRs in Lee et al., (2023), which were subsequently used by Tippett et al., (2024). More specifically, WR-A features spatial similarities to a Pacific Trough, WR-B and WR-D show warm and cool phases of a Pacific Ridge associated with ENSO, and WR-E is characterized by an Alaskan Ridge. WR-C features spatial similarities to a Greenland High as well. It is worth mentioning that our study focuses on a different region, a specific season and chooses a different k value, and there are thus noticeable differences. WR-A features two anomalous highs over the two coasts as opposed to one anomalous high over the central-CONUS. The anomalous low in WR-B is more pronounced than in Lee et al., (2023). The anomalous high in WR-C is wavelike unlike the Greenland high in Lee et al., (2023). The dipoles in WR-E are further south than they are in the Alaskan Ridge in Lee et al., (2023) …The favorable anomalies presented in WR-B agree with the Pacific Ridge findings in Tippett et al. (2024)."

The dependence of tornado activity as well as the dependence of CAPE and shear on month does not seem to have been accounted for in the analysis. In both cases anomalies are computed with respect to the April–July average. Using anomalies with respect to the April–July average means that the anomalies of quantities with a seasonal cycle will appear correlated but might actually be unrelated after accounting for seasonality.

*As stated in L115-116 of the original manuscript, the seasonal cycle is defined as the long-term mean at each grid-point for each calendar day. The anomalies in our study are calculated with respect to the daily climatology, instead of April-July average, as shown in Eq. 1, which helps remove the seasonal cycle.*

$$H'(d, y) = H(d, y) - \bar{H}(d) \qquad (1)$$

*where d denotes calendar day and y denotes year. This has been further clarified in the revised manuscript.* "Daily anomalies of MUCAPE, S06, and CP were calculated by removing the daily climatology on each calendar day, following Eq. 1."

*We chose not to normalize the tornado activity anomalies because we are working across one season where seasonality is not as big of an issue as it would be for a year-round study (as in Tippett et al. 2024). To maintain data consistency, we therefore chose not normalize CAPE or shear.*

*Regarding the dependence of tornado activity as well as the dependence of CAPE and shear on month, the coincidence of higher variance of tornado activity and CAPE in April-May might lead to an artificial correlation due to the seasonal cycle if we had used **monthly** mean data. However, this is not a concern for **daily** data because the strong coincidence of environmental condition*

*anomalies and tornado activity anomalies on a daily time scale likely indicates a physical relationship.*

Along with seasonality, ENSO may be another factor/alternative hypothesis to consider.

*We thank the reviewer for pointing this out. While ENSO was not explicitly explored in this manuscript, it is implied that the frequency of weather regime occurrence, CAPE, shear, and tornado activity may all be modulated by large-scale climate modes such as ENSO and MJO (Vigaud et al. 2018), with weather regimes serving as the intermediate piece between large-scale climate modes and tornado activity. Such relationships are outside the scope of the present study, but we have briefly discussed the potential role of low-frequency climate modes in the last section of the revised manuscript as it may help improve our prediction:*

*The following statement was added at the end of the conclusion:*

*"Furthermore, although not explored in this study, WRs and tornado activity may both be modulated by large-scale, low-frequency climate modes, with WRs potentially serving as the intermediate piece between large-scale climate modes and tornado activity, and the low-frequency modes may be important sources of predictability for the interannual variability of tornado activity."*

Abstract. "Our study highlights the potential application of WRs for better seasonal prediction of tornado activity." The authors' previous regime/tornado study examine subseasonal prediction of weekly regimes and found that forecast skill was lost at about Days 7–13. Is there evidence that these regimes are predictable on seasonal time scales?

*First, Miller et al. (2020) showed that the hybrid model has skill better than climatology out to Week 3. Second, with increasing forecast lead times, the information of the predictand will be less specific. Miller et al. focused on weekly mean tornado activity, but one may focus on seasonal mean tornado indices for seasonal prediction. Applying these regimes to seasonal prediction is our ongoing research, which shows promising results and we hope to publish in due time.*

The Weather regime methodology differs substantially from that used commonly in the literature. There are no explanations provided why. The weather regime classification method lacks standard diagnostics and assessments of robustness. The classification data is unavailable which means the classification cannot be applied by others to independent data and cannot be compared with other classifications (e.g., Lee et al., 2023 which provides the data)

*As explained above, what accounts for the "standard" regime classification is controversial. In particular, Falkena et al. (2020) argued against the use of either EOFs or time filtering on top of K-means clustering because K-means clustering reduces the dimensions and is a form of data filtration.*

*We have made our weather regime methodology code available at:*

*https://github.com/Matt0604/Kmeans*

*The ERA-5 and tornado report data are publicly available.*

*ERA-5:* https://doi.org/10.24381/cds.bd0915c6

*Tornado report:* https://www.spc.noaa.gov/wcm/#data

Line 106. "500H at 21 UTC was used to represent the daily circulation patterns." Previous weather regime classifications have used daily means and subsequently smoothed those in time, e.g., 10-day low-pass-filtered (Grams et al., 2017, 2020, Lee et al., 2023)

*The chosen time (2100 UTC) for 500H analysis represents a typical time of day when U.S. tornado outbreaks are ongoing (Cwik et al, 2022), thus potentially providing a more straightforward connection between WRs and tornado activity. In contrast, the use of 500H at times of day when tornado activity is much likely could result in misleading connections. Again, there is no standard data smoothing prior to K-means. Although Grams et al. (2017, 2020), Lee et al. (2023) applied a 10-day low-pass filter, some studies used a 5-day low-pass filter (Robertson and Ghil, 1999), and Falkena et al. (2020) cautioned against smoothing. Our analysis (Fig. R1) showed that applying a 5-day low-pass filtering does not qualitatively affect WR patterns.*

There is no EOF filtering which differs from previous work (Michelangeli et al., 1995, Grams et al., 2017, 2020, Lee et al., 2023)

*While many previous studies applied EOF and low-pass filtering prior to the clustering analysis, some studies chose to omit such procedures (Miller et al. 2020). In particular, Falkena et al. (2020; https://doi.org/10.1002/qj.3818) argued against the application of these procedures and advocated the use of the full field.*

Line 121. "the number of clusters was determined as five using the elbow method."

From the reference cited, the elbow method "is a visual method. The idea is that Start with K=2, and keep increasing it in each step by 1, calculating your clusters and the cost that comes with the training. At some value for K the cost drops dramatically, and after that it reaches a plateau when you increase it further. This is the K value you want." This is not really an objective method. Lee et al., 2023 apply four objective, data-driven methods for determining the best number of clusters, including the classifiability and reproducibility indices of Michelangeli et al. (1995).

*The choice of k is often somewhat subjective, because a metric does not always indicate an unambiguous optimal cluster number and different metrics may yield different optimal cluster numbers (Dorrington and Strommen 2020, https://doi.org/ 10.1029/2020GL087907). This is a known limitation of K-means. We tried k=4 given that Lee et al. (2023) chose k=4 in their analysis. As shown in Fig. R4, k =4 yields four of the same WR patterns as k=5 but misses WR B in the k = 5 analysis, which is spatially similar to WR-A in Miller et al. 2020 and the Pacific Ridge in Lee et*

*al. 2023. This regime is favorable for tornadoes and tornado outbreaks, so k=5 serves best for our purposes. Miller et al. 2020 also used k=5 based on the elbow method, which is a commonly used method for choosing k.*

There is no variance normalization to account for seasonality of variance (Grams et al., 2017 Lee et al., 2023). During the April–July period examined, Lee et al., (2023) found that the domain averaged Z500 std varied from 80 m in April to 50 m in July. Removing the daily climatology does not account for seasonality of variance.

*We agree that normalization is beneficial when examining weather regimes throughout the entire year, as it helps account for seasonality. However, most studies do not apply normalization when dealing with a specific season as opposed to the whole year. Since we are only looking at AMJJ, we chose not to normalize H500, which also helps maintain the consistency of our data analysis as explained before.*

Because k-means cluster analysis minimizes the total within-cluster variance, seasonality in the variance of the data means that clusters might be biased toward the later months of June and July when variance is small and consequently within-cluster variance is easier to minimize. Consequently the resulting cluster centroids are likely to be skewed toward patterns that best represent June/July variability at the expense of other months. And indeed, cluster A shows reduced activity which would be typical of the June/July period. This bias is potentially a serious flaw for the application here since US tornado activity is much higher in April–May than in June–July. In other words, tornado activity might be substantially higher in a particular weather regime simply because that regime is more frequent during calendar months when tornado activity is climatologically higher. Whether this is the case, and the association between tornado activity and regime frequency is simply due to their having similar seasonal cycles, is impossible to say because the authors have as far as I can see failed to provide any indication of the seasonality of cluster frequency or how the variance explained depends on month.

*We thank the reviewer for this comment, which brings up an interesting point. We agree that it is possible that cluster centroids might be skewed toward patterns that best represent June/July variability. However, given that tornado activity is climatologically higher in April and May, such skewness would disrupt the weather regime-tornado correlation instead of amplifying the correlation. In addition, even if a strong skewness exists and reflects in the seasonality of cluster frequency, it would not explain the link between the interannual variability of tornado activity and weather regime frequency as we demonstrated using our empirical model (Fig. 5 in the manuscript).*

*The seasonality of the WRs (Fig. R5) shows that WR-A and WR-B both occur more frequently later in the season. Given that WR-A is associated with reduced tornado activity and WR-B with enhanced tornado activity, this seasonality does not support the reviewer's speculation that "tornado activity might be substantially higher in a particular weather regime simply because that*

*regime is more frequent during calendar months when tornado activity is climatologically higher."*
*Additionally, although tornado activity decreases towards the end of July, it remains high during*
*June and July. As shown in Graber et al. (2024), the peak day for tornado days from 1960-1979*
*was June 14th.*

[Figure]

**Fig. R5:** Seasonality of WR frequency plotted as a 5-day running mean with dashed vertical lines
at the mean date.

Overall there are essentially no diagnostics of the WRs such as variance explained. Also
there is no assessment of how these regimes vary with large-scale modes of variability
such as ENSO (known to be important for tornado activity), NAO, etc.

*Similar to seasonality, because we are dealing with one season, the variance would be expected*
*to be more uniform and less spread out around the mean. This is especially true since we are*
*dealing with the warm season when the jet stream is less active. Accounting for variance would*
*certainly be more important if this was a year-round analysis where there would be more spread*
*in the data, as was done in Lee et al. (2023); Lee and Messori (2024) and Tippett et al. (2024).*

*By performing the low-pass filter and EOF filter analysis in figs R2 and R3, there is evidence that*
*the regimes we have created are capturing distinct modes of variability. In addition, Figure S4 in*
*the supplemental information illustrates the year-to-year and decadal variability of the 5 WRs in*

*addition to the persistence over time. The latter is an important point considering that multiple studies have shown that WR persistence may lead to more tornado outbreaks.*

*We agree that WR frequency may be modulated by large-scale climate modes such as ENSO and NAO which are relevant in tornado climatology, but how each WR varies with such modes goes beyond the scope of the current study.*

Line 134. The tornado probability anomalies fail to account for seasonality because they are with respect to the April–July frequency. Consequently, substantial anomalies may occur simply because some regime are more frequent during April–May rather than June–July. I see no exploration in the text of this hypothesis. This issue applies to environments as well reports.

*The tornado probability anomalies shown in Figure 2 were defined with respect to the daily climatology at each grid point, instead of with respect to the April–July average. We did not discuss this point because our data are properly deseasonalized and we do not think it would artificially amplify the WR-tornado correlations. As shown in Fig. R5, WR-B, which is associated with enhanced tornado activity, actually occurs more frequently during June-July, contrary to what the reviewer expected.*

*Line 138 now reads: "The TD probability anomalies ($P_a$) were calculated at each grid-point for each WR as follows:"*

Line 160. "These WRs have some spatial similarities to the year-round WRs found by Lee and Messori, (2024)." The more appropriate citation is Lee et al., (2023) which describes the classification in detail, and is not cited here. Also Lee et al., (2023) provide that classification data which means that authors here can make a more precision statement regarding the similarity of the classification. That is, with what frequency are the classifications the same. Also applying the diagnotic methods of Lee and Messori, (2024) to the regimes here would provide some evidence that regimes here are physically or dynamically meaningful.

*Lee et al. (2023) has been replaced by Lee and Messori (2024). We would like to point out that the region and season(s) examined in Lee et al. (2023) are different from those in our study, and some differences are thus expected.*

*Lines 97-100 now read: Year-round WRs (Lee et al. 2023) have also been used and found to have statistically significant relationships with tornado activity in all months except June-August (Tippett et al. 2024) although without any consideration of WR persistency."*

Line 165. "Composite anomalies of these parameters were calculated by subtracting the corresponding climatological mean." The same issue of using an inappropriate climatology applies to the MUCAPE and S06 anomalies, i.e., they will have seasonality both in their mean and variance. Because the April–July climatology is used, MUCAPE anomalies will tend to be positive in later months and negative in earlier months, and the opposite for S06. This means that any seasonality in the regime frequencies will project onto these anomalies, even if there is no relation when seasonality is taken into account. Moreover statistical significance tests used (e.g., Fig. 2) are inappropriate because they assume identically distributed but the variance of the data depends on month.

*Same as the other variable, the anomalies here are defined with respect to daily climatology, and seasonality is thus removed. We have made this clearer in the revised manuscript,*

"Daily anomalies of MUCAPE, S06, and CP were calculated by subtracting each calendar day's mean from every calendar day, following

$$H'(d,y) = H(d,y) - \bar{H}(d)$$

where $y$ is year, $d$ is calendar day, $H$ is the variable or parameter of consideration, and the overbar denotes the long-term mean."

Line 136. Tornado data from period 1960–2022 is used and it is claimed (line 132) regarding well-known report trends that "this trend is not reflected in TDs" (tornado days). However, Miller et al., (2020) with two of the same three authors conclude that the period 1990–2019 "represents a compromise between data set length and an allowance for a significant fraction of the reports to have occurred during the Next-Generation Radar era and thus have undergone some quality control." Moreover, Fig. 5a shows a very large, presumably secular shift in the number of tornado days, which is as large or larger than the year-to-year variability. The authors state later (line 294) that "modelled TDs are nearly out of phase with observations in the 1960s, when tornado reports are less reliable" which supports analysis on a shorter period.

*The tornado dataset is imperfect even in the current era, and therefore judicious choices must be made as a function of the particular application. The use of the term "compromise" by Miller et al. (2020) was not meant to suggest that tornado reports prior to 1990 were unusable, but rather that their particular application – hybrid S2S prediction–required relatively more certainty. Moreover, Miller et al. (2020)'s focus on hybrid prediction limited their maximum period length to 1990-2019 due to data availability in the ECMWF reforecasts. Since our study's focus was not prediction, we used the longer period length to match Graber et al. (2024) to look for ties between already observed tornado activity and seasonal weather regimes. The longer period also works*

*better with the results found by both Graber et al. (2024) and Brooks et al. (2014) since we are interested in providing physical explanations for the TD and TO trends that they found. Figs. 1 and 2 in Graber et al. 2024 demonstrate the "secular shift in the number of tornado days" in Figure 5a of the present study. April-July all shows large drop-offs in the 1980s, particularly April which has a few low outliers. One of the takeaways from that study was that the warm-season months were responsible for decreasing TD trend. The same shift appears in Figure 5a which is expected for the observations. Fig. S6 does provide the additional support that a shorter training period would be ideal for prediction studies, but the model performing adequately with the full period in spite of the report discrepancies supports the potential that seasonal weather regimes have to improve seasonal tornado prediction.*

Line 210. "may be possibly linked to tropical cyclones (Figs. 1e and 2e)." This is an interesting point and should be verified using data here https://www.spc.noaa.gov/exper/tctor/ and https://www.spc.noaa.gov/publications/edwards/tctor.xls

*Thank the reviewer for pointing out the dataset. Using this dataset, we found that there are 98 such TC-induced EF-1+ tornadoes during June and July but only 8 occurred during a WR-E. The result doesn't support our speculation, and the statement has been removed in the revised manuscript.*

Figs. 3 and 4 mention resampling to assess statistical significance without details. Depending on the design of the resampling procedure, the results may be incorrect if seasonality is not accounted for. For instance, in the case of a permutation test a possible way of taking seasonality into account is to compare tornado day frequency in say regime A with tornado day frequency on exactly the same calendar days when regime A did not occur.

*Significance in Figs. 3 & 4 was calculated using a Monte Carlo simulation test with 10000 resamples. The p-value was then calculated based on the proportion of simulations that were more extreme than the observation. More information has been added in the revised manuscript.*

*Lines 145-148 now read: "A Monte Carlo simulation test with 10000 resamples was used to test for significance of the anomalies. The number of WR-i days was multiplied by the climatological mean TD probability to get an expected number of tornado days. The p-value was calculated based on the proportion of simulations that were more extreme than the observations."*

*As explained before, the data are deseasonalized by removing the daily climatology, and we do not think seasonality of variance would induce an artificial link between WRs and tornado activity. To further address the reviewer's concern, we replotted Fig 4 for April-May and June-July (Fig R6), separately. Keep in mind that while tornado days still occur at a high rate in June and July,*

*tornado outbreaks peak during April and May, so the June-July Tornado outbreaks panel has a low sample size. The significant anomalies shown in Fig. 4 remain the same sign (except for persistent WR-E in June-July) with quantitative differences.*

[Figure]

**Fig. R6:** Tornado probability anomaly plots for persistent and nonpersistent TDs/TOs in April-May and June-July.

Fig. S4 "WR TD time series" it is unclear what this quantity is. If it is the number of tornado days in that weather regimen, then of course, there are fewer tornado days in that weather regime in years when that weather regime is less frequent (the very high correspondence between the blue and green curves). This would be the case even when there is no relation between tornado days and weather regimes. That being the case, reporting the correlation coefficient does not seem informative and might confuse some readers. (I think my interpretation of the green curve is correct because it is different in panels S4a-e.)

*Yes, your interpretation of the blue and green curves in Fig. S4 is correct. The correlation is expectedly high as pointed out by the reviewer, but we feel this figure is useful as it demonstrates the strong variability of weather regimes and its influence on tornado activity.*

Line 290 and Fig. 5. "the empirical model fails to capture the observed decreasing trend or the decadal shift in the 1980s." The use of Spearman correlation here may obscure the extent to which empirical model fails to capture observed variability. A scatterplot would likely give a much more accurate and pessimistic picture. The association is stated to be statistically significant at the 5% level but visually is hard to see. Perhaps a bootstrap test might give a more credible assessment of statistical significance.

*The poor visual agreement is mainly due to the underestimated tornado variance represented by the empirical model. We now show the observed and predicted time series with two separate y-axes (Fig. R7) so that it is easier to visualize the year-to-year fluctuations that lead to the Spearman correlation.*

[Figure]

**Fig. R7:** (New Figure 5 in the manuscript) Empirically modeled TDs (blue with circles) per year overlaid with (a) observed TDs (red with crosses) and (b) detrended observed TDs (red with crosses) with spearman rank correlation coefficient (cc) and p-value; (c) empirically modeled (blue with circles) and observed (red with crosses) TOs per year with the spearman rank correlation coefficient and p-value.

Line 312 and also the abstract. "the empirical model captures the interannual variability of TDs reasonably well" this seems an overly generous description.

*As stated in line 302, it is a common limitation of statistical modeling that the model curve will underestimate the magnitude of the year-to-year variability. We acknowledge the evidence of this in Fig. 5b & c. However, given the significance of the relationship between the observations and the model, we stand by this statement.*

*We have revised our statement to be more specific: The TD time series estimated by the empirical model shows a significant rank correlation (above the 95% confidence level) with the observed time series but underestimates the observed variance.*

Conclusions. Line 343. "A year that includes a high number of WR-B days is likely to have an above average number of TDs and TOs." Is there analysis/figure in the manuscript that supports this statement?

*Figures 1-4, along with Figure S2, all implicitly show this. The availability of more favorable environmental conditions in WR-B in Figs. 1 & 2 supports the higher WR-B tornado probability anomalies seen in Figs. 3 & 4. Since the probability values are used to create the empirical model, it is implied that a year with several WR-B days is likely to have an above average number of TDs and TOs.*

Line 285. "The frequencies of persistent WRs also show changes across different multidecadal time periods (Fig. S4f" I really don't see any substantial changes in Fig. S4f and I question whether a t-test is suitable for a change in frequency, perhaps Fisher's exact test.

*It is worth pointing out that the values are all normalized by the number of years in each period, and each period contains at least 20 years. The Fisher's Exact test is a good test to use for very small sample sizes (such as n<20), and while our sample size here is not that high, this test would not be the most useful for this figure.*

**References**

Brooks, H. E., G. W. Carbin, and P. T. Marsh, 2014: Increased Variability of Tornado Occurrence in the United States. *Science*, **346**, 349–352, https://doi.org/10.1126/science.1257460.

Cwik, P., R. A. McPherson, M. B. Richman, and A. E. Mercer, 2022: Climatology of 500-hPa Geopotential Height Anomalies Associated with May Tornado Outbreaks in the United States. *Int. J. Climatol.*, **43**, 893–913, https://doi.org/10.1002/joc.7841.

Dorrington, J., and K. J. Strommen, 2020: Jet Speed Variability Obscures Euro-Atlantic Regime Structure. *Geophys. Res. Lett.*, **47**, https://doi.org/10.1029/2020GL087907.

Falkena, S. K. J., J. de Wiljes, A. Weisheimer, and T. G. Shepherd, 2020: Revisiting the identification of wintertime atmospheric circulation regimes in the Euro-Atlantic sector. *Q. J. R. Meteorol. Soc.*, **146**, 2801–2814, https://doi.org/10.1002/qj.3818.

Graber, M., R. J. Trapp, and Z. Wang, 2024: The Regionality and Seasonality of Tornado Trends in the United States. *Npj Clim. Atmospheric Sci.*, **7**, https://doi.org/10.1038/s41612-024-00698-y.

Grams, C. M., R. Beerli, S. Pfenninger, I. Staffell, and H. Wernli, 2017: Balancing Europe's Wind-power Output through spatial development informed by weather regimes. *Nat. Clim. Change*, **7**, 557–562, https://doi.org/10.1038/nclimate3338.

Grams, C. M., L. Ferranti, and L. Magnusson, 2020: How to make use of weather regimes in Extended-range Predictions for Europe. *ECMWF Newsl.*,.

Lee, S. H., and G. Messori, 2024: The Dynamical Footprint of Year-Round North American Weather Regimes. *Geophys. Res. Lett.*, **51**.

——, M. K. Tippett, and L. M. Polvani, 2023: A New Year-Round Weather Regime Classification for North America. *J. Clim.*, **36**, 7091–7108, https://doi.org/10.1175/JCLI-D-23-0214.1.

Miller, D., Z. Wang, R. J. Trapp, and D. S. Harnos, 2020: Hybrid Prediction of Weekly Tornado Activity Out to Week 3: Utilizing Weather Regimes. *Geophys. Res. Lett.*, **47**, https://doi.org/10.1029/2020GL087253.

Robertson, A. W., and M. Ghil, 1999: Large-Scale Weather Regimes and Local Climate over the Western United States. *J. Clim.*, **12**, 1796–1813, https://doi.org/10.1175/1520-0442(1999)012%3C1796:LSWRAL%3E2.0.CO;2.

——, N. Vigaud, J. Yuan, and M. K. Tippett, 2020: Toward Identifying Subseasonal Forecasts of Opportunity Using North American Weather Regimes. *Mon. Weather Rev.*, **148**, 1861–1875, https://doi.org/10.1175/MWR-D-19-0285.1.

Tippett, M. K., K. Malloy, and S. H. Lee, 2024: Modulation of U.S. Tornado Activity by year-round North American Weather Regimes. *Mon. Weather Rev.*, https://doi.org/10.1175/MWR-D-24-0016.1.

Vigaud, N., A. W. Robertson, and M. K. Tippett, 2018: Predictability of Recurrent Weather Regimes over North America during Winter from Submonthly Reforecasts. *Mon. Weather Rev.*, **146**, 2559–2577.

---

## Author Comment (AC3)

Manuscript ID: egusphere-2024-3216

*We appreciate the comprehensive and constructive questions and suggestions from the reviewers. Below are our responses in blue italics. Any necessary revisions made to our manuscript are also indicated below.*

Reviewer # 2

The authors focus on the characterization of weather regimes that favor tornado outbreaks, and the persistence of such regimes. While an interesting presentation, their work does tread a similar vein to a number of recent studies, which aren't fully acknowledged in the introduction. There are several points where it is difficult to follow the methodological tests taken, and the applications of statistical significance tests are not clear as written. A few of the figures also need to be adjusted for accessibility. Finally, while the authors claim that these results will help with seasonal prediction, I'd actually like to see a little more of a connection as to how they envisage this improving on the existing paradigm. While these issues exist, they are not overly burdensome, which leads me to recommend a set of minor revisions.

*Thank you for your constructive feedback. The seasonal prediction is an aspect that we are presently working on and are excited to report in due time.*

**Minor Comments:**

Introduction, First Paragraph: I'm not convinced the approach taken by the authors here really fits with the manuscript. Talking about trends and variability does not connect well with their primary focus on the occurrence of weather regimes favoring tornado outbreaks at least as presented. The authors could stand to connect their work to analysis of trends and identifying what the gaps may be, which would help the flow of the text.

*The first paragraph was meant to start the introduction into our motivation for going in this direction. Our previous study* (Graber et al. 2024) *found trends in tornado days and tornado outbreaks in the Great Plains and Southeast. The warm-season trend throughout the CONUS had a noticeable decreasing trend. We lack a physical explanation for why this is occurring though, and the WR approach is our attempt to provide a physical explanation for this decrease. The modeling is done as a means of being useful to key stakeholders that value the results we provide.*

Line 48: The literature cited here portrays an incorrect precedence of the understanding derived. Arguably, any such reference to ENSOs influence on tornadoes should include Cook and Schaefer (2008), which was the first study to show a wintertime signal, and a reference to Allen et al. (2015), which showed the first statistically robust connection to the environment in springtime, 500 hPa geopotential

height and inferred cyclonic track, and observations, and developed the first seasonal prediction algorithm – which is directly relevant to the analyses presented here.

*Thank you for making note of these important references from the literature. Both studies have been added into the revised version of the manuscript:*

*"Cook and Schaefer, (2008) examined winter tornado outbreaks in relation to the phase of the El Niño – Southern Oscillation (ENSO) and found that a La Nina phase favored tornadoes in the Southeast and a neutral phase favored tornadoes in the Great Plains. Allen et al., (2015) further found that La Niña (El Niño) years typically coincide with more (fewer) tornadoes in the spring across the central CONUS, and that the winter ENSO phase can be used to predict tornado frequency during the spring."*

Paragraph beginning Line 73: It would seem appropriate to reference that the European community has long used weather regimes to look at relationships with significant severe weather events. See Punge and Kunz (2016) and references therein.

*We have cited Grams et al. 2017 in the revised manuscript who we believe was the first study to investigate year-round weather regimes which were done in Europe.*

*"Lee et al., (2023) applied the year-round WR method (Grams et al., 2017) over North America and defined four year-round WRs."*

Paragraph beginning Line 62: It seems strange to me that the literature cited for weather regimes does not include Tippet et al. (2024). The subsequent references to this paper also seem to neglect that study considering variability.

*(Tippett et al. 2024) was referenced in line 99 to discuss their findings in the year-round weather regime analysis and how they did not have any significant results in June-August. However, they did not discuss persistence in their study, which ours does. This paper came out not long before we initially submitted this manuscript to WCD. Lee et al. 2023, from the same research group, was also cited. It also important to cite Grams et al. (2017) here, which we have now included, since this was the study to first introduce the concept of year-round WRs. Grams et al. (2017) did year-round WRs over Europe and Lee et al. (2023) did it over the CONUS.*

*"Lee et al., (2023) applied the year-round WR method (Grams et al., 2017) over North America and defined four year-round WRs. Tippett et al, (2024) identified statistically significant relationships between these year-round WRs and tornado activity in all months except June through August, but Tippett et al. (2024) made no consideration of WR persistency."*

Line 108: It is not clear what is meant by the authors by the statement 'and to avoid too many near-zero CP values in a daily mean', suggest clarifying.

*A daily max was used for convective precipitation (CP) because there may be many hours in a day where no CP is occurring, thus if we were using hourly data we would probably get very low*

*values for almost every day since many hours would be equal to 0 mm/hr. This way, days with an hour or two of significant convection are easier to pick out.*

"Daily maximum values of MUCAPE and CP at each grid point were used to represent the daily peak instability and in turn amplify the signal for days with the potential for significant convective storms."

Line 116: What mean long term trend was used for the detrending procedure exactly? Given you are interested in variability, to what extent is the trend removed influenced by the ending year?

*This has been made clearer in the revised manuscript:* "The 500H data were then detrended by removing the linear trend of the seasonal mean (AMJJ) 500H averaged over the entire Northern Hemisphere (Fig. S1). The detrending approach removed the positive trend of hemispheric mean 500H caused by climate change while preserving the spatial patterns and potential changes of WR frequency or persistence."

Paragraph beginning Line 154: It would seem more appropriate to compare to the weather regimes of Tippett et al. (2024), given that these are tornado focused.

*Both Lee et al. 2023 and Tippett et,al. 2024 use the same weather regimes, but Tippett et al. 2024 would be a better reference to put in this paragraph given that they specifically focus on tornadoes. Lee et al. 2023 was the paper that originally introduced the annual weather regimes.*

"Some WRs are similar to the year-round WRs in Lee et al., (2023), which were subsequently used by Tippett et al., (2024). More specifically, WR-A features spatial similarities to a Pacific Trough, WR-B and WR-D show warm and cool phases of a Pacific Ridge associated with ENSO, and WR-E is characterized by an Alaskan Ridge. WR-C features spatial similarities to a Greenland High as well. It is worth mentioning that our study focuses on a different region, a specific season and chooses a different k value, and there are thus noticeable differences. WR-A features two anomalous highs over the two coasts as opposed to one anomalous high over the central-CONUS. The anomalous low in WR-B is more pronounced than in Lee et al., (2023). The anomalous high in WR-C is wavelike unlike the Greenland high in Lee et al., (2023). The dipoles in WR-E are further south than they are in the Alaskan Ridge in Lee et al., (2023)."

Line 221: How is the significance t-test performed here, and how is it indicated on the diagram and for what specifically? This is unclear.

*A one-sample t-test was performed between the climatological CP and the WR CP (so each day that was assigned that specific WR) at each gridpoint to test whether the WR mean is significantly different than the climatology at each gridpoint. The null hypothesis is that the sample mean of the anomalies is equal to zero.*

*The figure caption now read:* "Significance is tested at each grid point using a one-sample t-test with the null hypothesis that the sample mean of the anomalies is equal to zero, and anomalies with p-values≤0.05 are regarded as significant."

Line 362: The extent to which the WR would help with skillful seasonal forecasts is an interesting one. Given that existing seasonal prediction models do not incorporate weather regimes, and offer skillful forecasts, particularly when climate variability is strong (e.g. Allen et al. 2015, Lepore et al. 2018), I would encourage the authors to discuss what advantages applying the WR approach would achieve over existing models, and how that may contribute to more skillful forecasts rather than the current abstract stateme'nt.

*The following statement has been added at the end of the conclusion:*

*"Furthermore, although not explored in this study, WRs and tornado activity may both be modulated by large-scale, low-frequency climate modes. WRs could potentially act as an intermediary between large-scale climate modes and tornado activity, while the low-frequency modes may be important sources of predictability for the interannual variability of tornado activity."*

*The importance of this statement is that WRs offer an intermediate between climate change and tornado activity, which is a difficult connection to make considering the large difference in spatial scale size. The forecasting and predictability piece is currently being investigated by our group and will be published in due time.*

Figures: There is some colorblindness suitability issues with the figures, particularly green and red overlapping contours. Please address this in the revision.

*Thank you to the reviewer for pointing this out. This has been addressed in the revised version of the manuscript. Figure 1 CAPE shaded anomalies are now in the seismic color palette and the S06 anomaly contours are in gold. Figure 2 CP anomalies are now in the viridis color palette, and the tornado day probability anomalies are now magenta and cyan, respectively, for positive and negative anomalies.*

References:

Allen, J. T., Tippett, M. K., & Sobel, A. H. (2015). Influence of the El Niño/Southern Oscillation on tornado and hail frequency in the United States. *Nature Geoscience*, *8*(4), 278-283.

Cook, A. R., & Schaefer, J. T. (2008). The relation of El Niño–Southern Oscillation (ENSO) to winter tornado outbreaks. *Monthly Weather Review*, *136*(8), 3121-3137.

Lepore, C., Tippett, M. K., & Allen, J. T. (2017). ENSO-based probabilistic forecasts of March–May US tornado and hail activity. *Geophysical Research Letters*, *44*(17), 9093-9101.

Punge, H. J., & Kunz, M. (2016). Hail observations and hailstorm characteristics in Europe: A review. *Atmospheric Research*, *176*, 159-184.

**Our References**

Allen, J. T., M. K. Tippett, and A. H. Sobel, 2015: Influence of the El Nino/Southern Oscillation on tornado and hail frequency in the United States. *Nat. Geosci.*, **8**, 278–283, https://doi.org/10.1038/ngeo2385.

Cook, A. R., and J. T. Schaefer, 2008: The Relation of El Nino-Southern Oscillation (ENSO) to Winter Tornado Outbreaks. *Mon. Weather Rev.*, **136**, 3121–3137, https://doi.org/10.1175/2007MWR2171.1.

Graber, M., R. J. Trapp, and Z. Wang, 2024: The Regionality and Seasonality of Tornado Trends in the United States. *Npj Clim. Atmospheric Sci.*, **7**, https://doi.org/10.1038/s41612-024-00698-y.

Grams, C. M., R. Beerli, S. Pfenninger, I. Staffell, and H. Wernli, 2017: Balancing Europe's Wind-power Output through spatial development informed by weather regimes. *Nat. Clim. Change*, **7**, 557–562, https://doi.org/10.1038/nclimate3338.

Lee, S. H., M. K. Tippett, and L. M. Polvani, 2023: A New Year-Round Weather Regime Classification for North America. *J. Clim.*, **36**, 7091–7108, https://doi.org/10.1175/JCLI-D-23-0214.1.

Punge, H. J., and M. Kunz, 2016: Hail Observations and Hailstorm Characteristics in Europe: A Review. *Atmospheric Res.*, **176–177**, 159–184, https://doi.org/10.1016/j.atmosres.2016.02.012.

Tippett, M. K., K. Malloy, and S. H. Lee, 2024: Modulation of U.S. Tornado Activity by year-round North American Weather Regimes. *Mon. Weather Rev.*, https://doi.org/10.1175/MWR-D-24-0016.1.

---

## Author Response (AR2)

Manuscript ID: egusphere-2024-3216

*We appreciate the editor giving us a chance to respond and make further edits. Below, the reviewer's comments are in black, and our responses are in blue.*

The authors have made only a few substantive changes to the manuscript. It seems to me that a fair amount of the authors' response is arguing why they don't need to respond to comments or simply not responding. In my view, the most egregious problem is that authors refused to either provide their classification data or to make quantitation comparisons: with previous work, including their own.

*We appreciate the reviewer for taking the time to review our revised manuscript. However, we respectively disagree with the statement that we "made only a few substantive changes" or refused to provide the data. We made many changes to the manuscript in response to the comments of both reviewers. When we chose to stand by our methodology, we provided justifications both from previous studies and additional new analysis.*

*Our methodology preference is supported by prior studies, which have indicated that some of reviewer 1's suggested changes may not be necessary given the specific objectives of our work. However, we appreciate the opportunity to further clarify and justify our work. We value the reviewer's insights and are willing to make modifications where necessary.*

*Firstly, the reviewer claims we made only a few substantive changes to the manuscript following the previous set of revisions. Below we outline the major changes that we made to the manuscript from the previous set of revisions:*

1) *Regarding the comparisons with previous work, we had added the following text in the last revision.*

*We added lines 48-54 in the last round of revision:* "For example, Cook and Schaefer, (2008) examined winter tornado outbreaks in relation to the phase of the El Niño – Southern Oscillation (ENSO) and found that a La Niña phase favored tornadoes in the Southeast and a neutral phase favored tornadoes in the Great Plains. Allen et al., (2015) further found that La Niña years typically coincide with more tornadoes in the spring and El Niño years with fewer tornadoes across the central CONUS, and that the winter ENSO phase can be used to predict tornado frequency during the spring."

*The prior text was added to give better precedence to prior work done on ENSO in relation to tornado outbreaks.*

*We added lines 89-93 in the last round of revision:* "Lee et al., (2023) applied the year-round WR method (Grams et al., 2017) over North America and defined four year-round WRs. Tippett et al., (2024) identified statistically significant relationships between these year-round WRs and

tornado activity in all months except June through August, but Tippett et al. (2024) made no consideration of WR persistency."

*The prior text was added to give more credit to WR work done by Lee et al. (2023) and Tippett et al. (2024) given their work done involving annual WRs and tornado activity.*

*2) More information was provided about our methodology.*

*We added lines 118-121 in the last round of revision:* "Daily anomalies of a variable were calculated by subtracting each calendar day's mean from every calendar day, following:

$$H'(d, y) = H(d, y) - \overline{H}(d) \tag{1}$$

where $y$ is year, $d$ is calendar day, $H$ is the variable or parameter of consideration, and the overbar denotes the long-term mean."

*The prior text was added in the methodology to describe the anomalies of MUCAPE, S06, and CP in figures 1 and 2. It was an initial concern by reviewer # 1 of whether the anomalies were with respect to the seasonal or daily mean. We made it clearer that these anomalies are with respect to the daily mean, and the seasonal cycle is thus removed.*

*We added lines 144-152 in the last round of revision:* "While many previous studies applied a low-pass filter or/and EOF dimension reduction prior to K-means clustering analysis (e.g., Grams et al., 2020; Lee et al., 2023; Lee and Messori, 2024; Robertson et al., 2020), Falkena et al., (2020) cautioned against the use of either EOFs or time filtering on top of K-means clustering. Our analysis shows that the application of the 5-day running mean or EOF dimension reduction prior to K-means does not qualitatively affect the regime patterns or the regime frequencies (Figs. S2-S4). We thus chose to use the simplest procedures for regime classification. Additionally, unlike Tippett et al., (2024), 500H anomalies are not normalized as we focus on one season, so seasonality is not as much of a concern."

*The prior text was added to the manuscript to better explain and defend our weather regime methodology. It is true that Lee et al. (2023) and subsequent studies applied a low-pass filter and/or EOF dimension reduction to their 500H anomalies prior to K-means clustering analysis, but Falkena et al. (2020) made a point that this is not necessary on top of K-means clustering. In our previous response, we made WR spatial structures which applied both a 5-day low pass filter and an EOF filter prior to K-means clustering, and it showed very little change. Since our work focuses on one season, the seasonality effect is not as a big issue as in a year-round analysis. This text has since been removed in the latest version to be consistent with our updated methodology.*

*We added lines 163-166 from the first revision state: "A Monte Carlo simulation test with 10000 resamples was used to test for significance of the anomalies. The number of WR-i days was multiplied by the climatological mean TD probability to get an expected number of tornado days. The p-value was calculated based on the proportion of simulations that were more extreme than the observations."*

*The prior text was added to provide a more detailed description of the significance testing done in figures 3 and 4.*

*3) More information was added in the results section on WR comparison.*

*We added lines 203-213 in the last round of revision: "Some WRs are similar to the year-round WRs in Lee et al., (2023), which were subsequently used by Tippett et al., (2024). More specifically, WR-A features spatial similarities to a Pacific Trough, WR-B and WR-D show warm and cool phases of a Pacific Ridge associated with ENSO, and WR-E is characterized by an Alaskan Ridge. WR-C features spatial similarities to a Greenland High as well. It is worth mentioning that our study focuses on a different region, a specific season and chooses a different k value, and there are thus noticeable differences. WR-A features two anomalous highs over the two coasts as opposed to one anomalous high over the central-CONUS. The anomalous low in WR-B is more pronounced than in Lee et al., (2023). The anomalous high in WR-C is wavelike unlike the Greenland high in Lee et al., (2023). The dipoles in WR-E are further south than they are in the Alaskan Ridge in Lee et al., (2023)."*

*The prior text was added to provide more detailed descriptions on the 5 WRs and relate them to work done by Lee et al. (2023) and Tippett et al. (2024). We discuss spatial similarities between our work and theirs. This text has since been modified to accommodate qualitative comparisons from multiple studies.*

*4) Additional discussion was added on the possibility that WRs and tornado activity have sources of predictability from large-scale, low-frequency climate modes.*

*We added lines 400-406 in the last round of revision: "Furthermore, although not explored in this study, WRs and tornado activity may both be modulated by large-scale, low-frequency climate modes (Cook and Schaefer 2008; Lee et al. 2023; Niloufar et al. 2021; Tippett et al. 2024; Vigaud et al. 2018). Given the potential predictability of WRs (Straus et al. 2007), they may act as an intermediary between large-scale climate modes and tornado activity, while the low-frequency modes may be important sources of predictability for the interannual variability of tornado activity."*

*The prior text was added to acknowledge that the WRs and tornado activity may be modulated by large-scale, low-frequency climate modes, but it went beyond the focus of this studies to investigate the sources of predictability. This is something that we are presently exploring.*

_**Below we summarize the major changes made to the new manuscript**_ _in response to the reviewer 1's comments on the revised manuscript. More information is provided in our replies to the specific comments._

1) _Daily mean 500H anomalies with 5-day low-pass filtering and EOF dimension reduction are used._
2) _The Davies-Bouldin Index_ (Davies and Bouldin 1979) _is included as an additional cluster identification method to support the choice of the optimal cluster number. This is one of the four methods described in Lee et al. (2023)._

3) _More qualitative WR comparisons with previous studies are added._

_**Below we address reviewer # 1's specific points.**_

Author responses are in quotes.

1. Lack of Variance Normalization. I noted that previous work showed that the variance of 500 hPa height anomalies varies significantly across months (April variance is higher than July), and this is a reason for normalization.

"Whether the data should be normalized is an interesting question for debate." A more responsive response would be to check whether variance normalization impacts the results.

"the variance would be expected to be more uniform and less spread out around the mean." In original review I noted that Lee et al. (2023) shows this is not the case. Also, I don't know what it means for the variance to be spread around the mean.

_One potential issue with the methodology in Lee et al. (2023) is that the anomalies are normalized by an area-averaged standard deviation with a cosine-latitude weighting. Since the standard deviation of geopotential height increases with latitude and the cosine-latitude weighting decreases with latitude, such normalization inflates the anomalies at higher latitudes, where the actual standard deviation is higher than the areal-averaged standard deviation. It contributes to the strong high-latitude loading of the WRs in Lee et al. (2023). We thus choose not to do the normalization._

_We also want to point out that many studies on WRs for a specific season do not normalize the anomalies (Miller et al. 2020; Robertson and Ghil 1999; Vigaud et al. 2018a; Zhang et al. 2024). In particular, Zhang et al. (2024) used WRs to study west coast wildfires during June-October, and no variance normalization was done. The 5 CONUS WRs that we have identified are similar to WRs identified in Zhang et al. (2024)._

*We justified our methodology in lines 133-137 in the new revision: "*Although geopotential height anomalies were normalized prior to the K-means clustering in the year-round WR analysis by Tippett et al. (2024) and Lee et al. (2023), 500H anomalies are not normalized in this study because we focus on one season, which is consistent with many previous studies (Miller et al., 2020; Robertson and Ghil, 1999; Vigaud et al., 2018a; Zhang et al., 2024)."

2. My first review: The new regime classification is not compared with previous ones from the same authors for April and May and with year-round regime classifications from Lee et al., (2023) [data is in Zenodo for download]. Making connections to previous work would increase the value of the current work. The classification data (data needed to classify independent data and classification of the days in the study) should be provided. I also asked: with what frequency are the classifications the same. No response.

A responsive response would have been to make a quantitative comparison of new weather regime classification with the previous ones. The data is available for that, and authors chose not to. They also have failed to provide the classification data. Code and a link to ERA5 data is not really very helpful.

*Qualitative comparisons were made to Lee et al. (2023) and Miller et al (2020) in the last revision, but we don't think a quantitative comparison is necessary because i) our study and those studies focus on different seasons and/or geographic regions; ii) there are many previous studies on weather regimes, and a quantitative comparison with just two of them does not provide much added value.*

*On the other hand, additional comparisons are made in the revised manuscript to a recently published paper, Zhang et al. (2024). This study used seasonal WRs from June-October to study wildfires on the west coast, using a total of 5 clusters. Our WR spatial structures are very similar to theirs with only minor differences in long-term frequency.*

*Lines 184-191 now state: "*The WR spatial structures closely resemble the WRs in Zhang et al. (2024) and Miller et al. (2020). WR-A is also similar to the Alaskan ridge pattern in Lee et al., (2023) and Tippett et al., (2024), and WR-D is similar to their Pacific ridge pattern. However, since our study focuses on a different region and a specific season, and is based on a different number of clusters, there are noticeable differences. In particular, the WRs in Lee et al. (2023) have a stronger loading in higher latitudes, probably partly because they normalized geopotential height anomalies by the area-averaged standard deviations with a cosine-latitude weighting, a procedure we choose to exclude."*

*Also, we respectfully disagree with the reviewer's comment that "Code and a link to ERA5 data is not really very helpful." We believe that sharing code ensures transparency in how the results are produced and enables users to easily reproduce our findings using the publicly available ERA5 data. Nevertheless, the WR label and spatial pattern data have been also made available in the GitHub site now.*

3. "The 'once-per-day snapshot' approach was pursued because the chosen time (2100 UTC) of

500H represents a typical time of day when U.S. tornado outbreaks are ongoing." This makes sense for tornadoes but not for weather regimes which are supposed to be persistent features.

*We have made this change and used daily anomalies in the revised manuscript, although using either a once-per-day snapshot or daily means does not qualitatively change any of the results or the physical interpretation. As shown in the two figures below (Figs. R1-R2), the analyses using the 21z vs daily mean show minor changes in long-term frequency of each WR, but the spatial structures remain the same and the modeling results are comparable.*

*Lines 112-114 now read:* "Data from the ERA-5 reanalysis (Hersbach et al., 2020) were analyzed over the CONUS $[24 - 55^{o}\ \text{N}, 130 - 60^{o}\ \text{W}]$ at the native $0.25^{o}$ latitude $\times$ $0.25^{o}$ longitude resolution. This includes daily mean 500 hPa geopotential heights (500H)."

[Figure]

**Figure R1:** WRs and empirical modeling using 21z snapshot anomalies (from the original manuscript) and 5-day low-pass filtering.

[Figure]

**Figure R2:** WRs and empirical modeling using daily mean anomalies and 5-day low-pass filtering.

4. Given that previous work with weather regimens has reported relations with ENSO and other modes of variability, the refusal to do so here is hard to understand. This added text is not particularly helpful in that regard.

"Furthermore, although not explored in this study, WRs and tornado activity may both be modulated by large-scale, low-frequency climate modes, with WRs potentially serving as the intermediate piece between large-scale climate modes and tornado activity, and the low-frequency modes may be important sources of predictability for the interannual variability of tornado activity." This is speculation, and fails to acknowledge previous work that has already considered these issues.

*Since our study focuses on the link between WRs and tornado activity, exploring the relations between WRs and various climate modes (including ENSO), or the sources of predictability, goes beyond the scope of this work. Our statement that the reviewer cited above is a reasonable hypothesis based on previous studies and a focus of our other ongoing study, which we plan to report in an upcoming manuscript. We believe that the current results, on their own, are worthy of publication.*

*Additionally, previous studies were cited about the impacts of various climate modes on tornado activity in the introduction of the manuscript, and more citations are now added in the revised manuscript:*

*Lines 400-406 now state:* "Furthermore, although not explored in this study, WRs and tornado activity may both be modulated by large-scale, low-frequency climate modes (Cook and Schaefer 2008; Lee et al. 2023; Niloufar et al. 2021; Tippett et al. 2024; Vigaud et al. 2018). Given the potential predictability of WRs (Straus et al. 2007), they may act as an intermediary

between large-scale climate modes and tornado activity, while the low-frequency modes may be important sources of predictability for the interannual variability of tornado activity."

5. In the first review I questioned whether the manuscript supported the abstract statement "Our study highlights the potential application of WRs for better seasonal prediction of tornado activity." In particular, I asked: Is there evidence that these regimes are predictable on seasonal time scales? The statement remains in the abstract, and the question of seasonal prediction was left unanswered by their response which talks about weekly timescales.

"First, Miller et al. (2020) showed that the hybrid model has skill better than climatology out to Week 3. Second, with increasing forecast lead times, the information of the predictand will be less specific. Miller et al. focused on weekly mean tornado activity, but one may focus on seasonal mean tornado indices for seasonal prediction. Applying these regimes to seasonal prediction is our ongoing research, which shows promising results and we hope to publish in due time."

*The AGCM simulations carried out by* Straus et al. (2007)*, forced by observed SST and sea ice, implies the predictability of weather regimes, which is also supported by our ongoing research. We prefer to keep this statement in the abstract as it reflects what motivates this study and where we anticipate our future study to go from here, but we have added more discussion in the last section of the manuscript, along with a new reference:*

*Lines 390-395 now state: "*Since the empirical model used WR frequency derived from the ERA reanalysis, its predictive skill can be considered an upper bound for this empirical prediction framework, assuming the perfect knowledge of WR frequencies. The atmospheric general circulation model simulations by Straus et al. (2007), which were forced by observed SST and sea ice, suggested the predictability of WRs, thus indicating the potential value of this approach."

*lines 403-406 now state: "*Given the potential predictability of WRs (Straus et al. 2007), they may act as an intermediary between large-scale climate modes and tornado activity, while the low-frequency modes may be important sources of predictability for the interannual variability of tornado activity."

*The hybrid model results shown in Miller et al. (2020) showed promising results on the subseasonal timescale, and now our study is attempting to expand to the seasonal timescale. A hybrid model for seasonal tornado prediction has not yet been investigated, hence why the potential we refer to in the manuscript remains an open question. Our study provides evidence that this is a project worth to continue exploring, and we will present more results in due time.*

6. In the first review I said: The weather regime classification method here lacks standard diagnostics and assessments of robustness. There is no mention in their response of diagnostics or tests for robustness. There are multiple common diagnostics for weather regime calculations that have been used previously.

"As explained above, what accounts for the "standard" regime classification is controversial. In particular, Falkena et al. (2020) argued against the use of either EOFs or time filtering on top of K-means clustering because K-means clustering reduces the dimensions and is a form of data filtration. We have made our weather regime methodology code available"

*We have added the Davies-Bouldin Index, discussed in detail under point 7, as evidence that we have chosen the correct # of WRs.*

*We now apply a 5-day low-pass filter and use the first 8 eofs for EOF dimension reduction as requested by the reviewer. All figures and in-text numbers have been changed to address the new WRs, but the physical interpretation of our results remains largely the same.*

*Lines 137-140 state: "A 5-day low-pass filter was applied to 500H anomalies, and the leading 8 EOFs, accounting for ~90% of the variance, were retained in the EOF dimension reduction; such pre-processing does not qualitatively affect the regime patterns or the regime frequencies (Figs. S2-S4) but does facilitate comparison with previous studies."*

*Regarding robustness, we tested various analysis approaches for data pre-processing as requested by the reviewer, and they all yield very similar results, which supports the robustness of our analysis. In addition, the relationship between WRs and environmental parameters, such as CAPE, VWS, and precipitation, corresponds well with the WR-tornado link. Such process-level analysis helps further confirm the robustness of our results.*

*Figs R1-R5 show that the WRs and modeling results undergo little change with whatever methodology is chosen. Fig. R5 shows the WRs that will be used in the new version of the manuscript. All statistics in the manuscript text have been changed to go with the results of the Fig. R5 WRs.*

[Figure]

**Fig. R3:** 21Z WRs without any filtering.

[Figure]

**Fig. R4:** 21z WRs using a 5-day low-pass filter and the first 8 eofs which account for ~90% of the variance.

[Figure]

**Fig. R5:** Daily Mean WRs using a 5-day low-pass filter and the first 8 eofs which account for ~90% of the variance.

7. The "elbow method" is not an objective method for choosing the truncation. I noted that: Lee et al., (2023) apply four objective, data-driven methods for determining the best number of clusters, including the classifiability and reproducibility indices of Michelangeli et al. (1995). Again the authors have chosen to ignore that comment and to apply no such objective diagnostics to their classification.

*We have added the Davies-Bouldin Index (Fig. R6), which is one of the classification methods in Lee et al. (2023), and this shows that 5 clusters are indeed the optimal number of clusters. As stated in Lee et al. (2023) and explained fully in Davies and Bouldin, (1979), this method measures the average similarity between clusters, and when the index value reaches a minimum point, adding more clusters makes it harder to distinguish the patterns in each WR. For the purposes of our study, 5 clusters are an appropriate number and goes with previous work (Miller et al. 2020, Zhang et al. 2024). The plot shown is for WRs using daily mean 500H anomalies, as now done in the manuscript.*

*Lines 142-143 now mention use of both the elbow method and the Davies-Bouldin Index.*

[Figure]

**Fig. R6:** Davies-Bouldin Index for each number of clusters

8. In my first review I said: Overall there are essentially no diagnostics of the WRs such as variance explained. There is no indication in the response that the authors have added any diagnostics including variance explained.

*In our original analysis, since total anomaly field is used, the WR explains 100% of the variance. We have now employed the EOF dimension reduction in the revised manuscript, and the first 8 eofs account for ~90% of the total variance.*

9. In my first review I questioned whether their statistical tests accounted for seasonality. Again I see no response to this question. I assume that seasonality was not accounted for, despite the seasonality of the data. Removal of the mean is inadequate in this regard because sampling variability depends on variability.

*One potential issue with the methodology in Lee et al. (2023) is that the anomalies are normalized by an area-averaged standard deviation with a cosine-latitude weighting. Since the standard deviation of geopotential height increases with latitude and the cosine-latitude weighting decreases with latitude, such normalization inflates the anomalies at higher latitudes, where the actual standard deviation is higher than the areal-averaged standard deviation. It contributes to the high-latitude strong loading of the WRs in Lee et al. (2023). We thus choose not to do the normalization.*

*We also want to point out that many studies on WRs for a specific season do not normalize the anomalies (Miller et al. 2020; Robertson and Ghil 1999; Vigaud et al. 2018a; Zhang et al. 2024). In particular, Zhang et al. (2024) used WRs to study west coast wildfires during June-October, and no variance normalization was done. The 5 CONUS WRs that we have identified are similar to WRs identified in Zhang et al. (2024).*

10. I asked whether the correlation reported in the original S4 figure was misleading because there are fewer tornado days in a particular weather regime in precisely those years when that weather regime itself is less frequent. The response seems to be yes, it is misleading but "we feel this figure is useful as it demonstrates the strong variability of weather regimes and its influence on tornado activity." This statement is patently false because the correlation mainly reflects the regime frequency, not the association of the regime with tornado activity, a fact that even the authors acknowledge when they say in the response (but not in the manuscript) "the correlation is expectedly high."

*The main purpose of Figure S4, now Figure S8, was to show the strong interannual and decadal variability of WR frequencies. We added in the tornado day observations curves as a supplement. The TD observation curve and subsequent correlations have now been removed from this figure to address the reviewer's concern. The plot now includes a trend line of the seasonal frequency.*

Lines 314-319 now state: "In this section, we further quantify the link between WRs and tornado activity. WR frequencies demonstrate strong interannual and decadal variability (Fig. S8a-e). In particular, WR-A exhibits a frequency increase during the 1980s coinciding with the steepest decrease in TDs (Brooks et al., 2014; Graber et al., 2024). The increase in seasonal frequency in WR-A is consistent with the spatially similar ridge-trough-ridge WR in Zhang et al., (2024). The frequencies of persistent WRs also show changes across different multidecadal time periods (Fig. S8f)."

***Additional minor edits were made to the manuscript that were not in response to the reviewer:***

*Figure captions have been modified to fully reflect what each figure is showing including statements on use of t-tests for significance testing, 2º (latitude) x 2º (longitude) uniform filtering for the convective anomalies, and a gaussian filter for the TD probability anomalies in Figure 2.*

*This includes an added statement to the methodology on the uniform filtering of convective parameter anomalies.*

*Lines 121-123 now state:* "A 2º (latitude) x 2º (longitude) uniform filter was applied to MUCAPE, CP, and S06 anomalies to coarsen the data and were tested for significance using a one-sample, two-sided t-test."

*Lines 225-227 now state:* "Here TD probability anomalies are evaluated following Eq. 2 with respect to $P_c$ at each grid point and then smoothed using a scipy gaussian filter with sigma 6. Such smoothing has removed some small-scale anomalies but retained the large-scale patterns."

*Figures 1 and 2 have been modified to make them easier to read.*

*Figure 6, change in MUCAPE and CIN was removed from the main manuscript and was placed in the SI. The new figure includes smaller changes in S06 superimposed on the CAPE plots. A brief description of the figure is still included in the manuscript to explain a possible limitation of the model.*

*Lines 351-355 now state:* "Additionally, convective inhibition (CIN) increases in the Southeast and Midwest for WR-B (Fig. S11h) and in the central-CONUS for WR-C (Fig. S11i) from P1 to P3. Further analysis reveals a lower TD probability for all WRs in P3 than in P1 (not shown), consistent with the negative trend of TDs (Graber et al. 2024)."

References:

Allen, J. T., M. K. Tippett, and A. H. Sobel, 2015: Influence of the El Nino/Southern Oscillation on tornado and hail frequency in the United States. *Nat. Geosci.*, **8**, 278–283, https://doi.org/10.1038/ngeo2385.

Brooks, H. E., G. W. Carbin, and P. T. Marsh, 2014: Increased Variability of Tornado Occurrence in the United States. *Science*, **346**, 349–352, https://doi.org/10.1126/science.1257460.

Cook, A. R., and J. T. Schaefer, 2008: The Relation of El Nino-Southern Oscillation (ENSO) to Winter Tornado Outbreaks. *Mon. Weather Rev.*, **136**, 3121–3137, https://doi.org/10.1175/2007MWR2171.1.

Davies, D. L., and D. W. Bouldin, 1979: A Cluster Separation Measure. *IEEE Trans. Pattern Anal. Mach. Intelligience*, **PAMI-1**, 224–227, https://doi.org/10.1109/TPAMI.1979.4766909.

Falkena, S. K. J., J. de Wiljes, A. Weisheimer, and T. G. Shepherd, 2020: Revisiting the identification of wintertime atmospheric circulation regimes in the Euro-Atlantic sector. *Q. J. R. Meteorol. Soc.*, **146**, 2801–2814, https://doi.org/10.1002/qj.3818.

Graber, M., R. J. Trapp, and Z. Wang, 2024: The Regionality and Seasonality of Tornado Trends in the United States. *Npj Clim. Atmospheric Sci.*, **7**, https://doi.org/10.1038/s41612-024-00698-y.

Grams, C. M., R. Beerli, S. Pfenninger, I. Staffell, and H. Wernli, 2017: Balancing Europe's Wind-power Output through spatial development informed by weather regimes. *Nat. Clim. Change*, **7**, 557–562, https://doi.org/10.1038/nclimate3338.

Grams, C. M., L. Ferranti, and L. Magnusson, 2020: How to make use of weather regimes in Extended-range Predictions for Europe. *ECMWF Newsl.*

Hersbach, H., and Coauthors, 2020: The ERA5 global reanalysis. *Q. J. R. Meteorol. Soc.*, **146**, 1999–2049, https://doi.org/10.1002/qj.3803.

Lee, S. H., and G. Messori, 2024: The Dynamical Footprint of Year-Round North American Weather Regimes. *Geophys. Res. Lett.*, **51**.

——, M. K. Tippett, and L. M. Polvani, 2023: A New Year-Round Weather Regime Classification for North America. *J. Clim.*, **36**, 7091–7108, https://doi.org/10.1175/JCLI-D-23-0214.1.

Miller, D., Z. Wang, R. J. Trapp, and D. S. Harnos, 2020: Hybrid Prediction of Weekly Tornado Activity Out to Week 3: Utilizing Weather Regimes. *Geophys. Res. Lett.*, **47**, https://doi.org/10.1029/2020GL087253.

Niloufar, N., N. Devineni, V. Were, and R. Khanbilvardi, 2021: Explaining the Trends and Variability in the United States Tornado Records using Climate Teleconnections and Shifts in Observational Practices. *Sci. Rep.*, **11**, https://doi.org/10.1038/s41598-021-81143-5.

Robertson, A. W., and M. Ghil, 1999: Large-Scale Weather Regimes and Local Climate over the Western United States. *J. Clim.*, **12**, 1796–1813, https://doi.org/10.1175/1520-0442(1999)012%3C1796:LSWRAL%3E2.0.CO;2.

——, N. Vigaud, J. Yuan, and M. K. Tippett, 2020: Toward Identifying Subseasonal Forecasts of Opportunity Using North American Weather Regimes. *Mon. Weather Rev.*, **148**, 1861–1875, https://doi.org/10.1175/MWR-D-19-0285.1.

Straus, D. M., S. Corti, and F. Molteni, 2007: Circulation Regimes: Chaotic Variability versus SST-Force Predictability. *J. Clim.*, **20**, 2251–2272, https://doi.org/10.1175/JCLI4070.1.

Tippett, M. K., K. Malloy, and S. H. Lee, 2024: Modulation of U.S. Tornado Activity by year-round North American Weather Regimes. *Mon. Weather Rev.*, https://doi.org/10.1175/MWR-D-24-0016.1.

Vigaud, N., A. W. Robertson, and M. K. Tippett, 2018: Predictability of Recurrent Weather Regimes over North America during Winter from Submonthly Reforecasts. *Mon. Weather Rev.*, **146**, 2559–2577.

Zhang, W., S. S-Y Wang, Y. Chikamoto, R. Gillies, M. LaPlante, and V. Hari, 2024: A Weather Pattern Responsible for Increasing wildfires in the western United States. *Environ. Res. Lett.*, **20**, https://doi.org/10.1088/1748-9326/ad928f.